# EXP-BENCH: CAN AI CONDUCT AI RESEARCH EXPERIMENTS?

**Patrick Tser Jern Kon**[1]**, Qiuyi Ding**[1]**, Jiachen Liu**[1]**, Xinyi Zhu**[1]**, Jingjia Peng**[1]
**Jiarong Xing**[2,4]**, Yibo Huang**[1]**, Yiming Qiu**[1,4]**, Jayanth Srinivasa**[3]**, Myungjin Lee**[3]
**Mosharaf Chowdhury**[1]**, Matei Zaharia**[4]**, Ang Chen**[1]
[1]University of Michigan  [2]Rice University  [3]Cisco Research  [4]UC Berkeley

## ABSTRACT

Automating AI research holds immense potential for accelerating scientific progress, yet current AI agents struggle with the complexities of rigorous, end-to-end experimentation. We introduce EXP-Bench, a novel benchmark designed to systematically evaluate AI agents on complete research experiments sourced from influential AI publications. Given a research question and incomplete starter code, EXP-Bench challenges AI agents to formulate hypotheses, design and implement experimental procedures, execute them, and analyze results. To enable the creation of such intricate and authentic tasks with high-fidelity, we design a semi-autonomous pipeline to extract and structure crucial experimental details from these research papers and their associated open-source code. With the pipeline, EXP-Bench curated 461 AI research tasks from 51 top-tier AI research papers. Evaluations of leading LLM-based agents, such as OpenHands and IterativeAgent on EXP-Bench demonstrate partial capabilities: while scores on individual experimental aspects such as design or implementation correctness occasionally reach 20–35%, the success rate for complete, executable experiments was a mere 0.5%. By identifying these bottlenecks and providing realistic step-by-step experiment procedures, EXP-Bench serves as a vital tool for future AI agents to improve targeted research components and agent planning ability. EXP-Bench is available at `https://github.com/Just-Curieous/Curie/tree/main/benchmark/exp_bench`.

## 1 INTRODUCTION

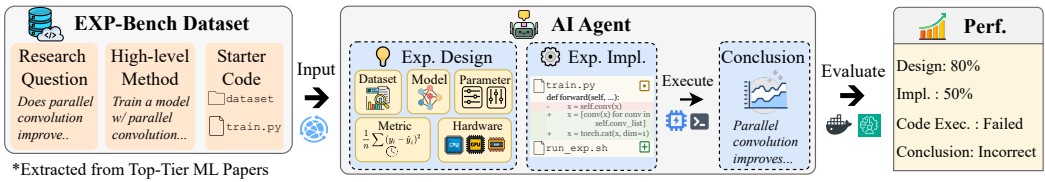

Figure 1: EXP-Bench evaluates AI agents on research experiment tasks extracted semi-autonomously from peer-reviewed AI papers. Given a research question, a high-level method description, and starter code, agents are tasked with designing, implementing, and executing complete experiments. Performance is validated through ground-truth comparisons and implementation execution.

Automating AI research stands as a cornerstone for accelerating the development of advanced intelligence and human progress. Unlike disciplines that require extensive physical interaction, AI research is inherently digital, rendering it particularly amenable to automation by Large Language Model (LLM)-driven AI agents. Recent work has demonstrated that these agents demonstrate nascent capabilities in tasks like literature synthesis (Elicit, 2025), hypothesis generation (Weng et al., 2025) and code generation (Li et al., 2022). However, empirical AI research requires rigorous end-to-end experimentation, which goes beyond these individual tasks.

To realize the vision of agents conducting holistic AI research, a rigorous benchmark is needed—one that evaluates and guides agents through the full experimentation pipeline step by step. We present EXP-Bench, a benchmark designed to comprehensively assess an AI agent's ability to carry out end-to-end research experiments. As illustrated in Fig. 1, EXP-Bench challenges agents with tasks sourced from influential, peer-reviewed AI publications (e.g., NeurIPS, ICLR) along with their open-source implementations. These papers reflect already-completed, peer-validated research and serve as concrete exemplars of full experimental workflows. By exposing agents to such tasks, we test their ability to conduct established scientific procedures grounded in real-world AI experimentation. For each task, an agent is provided with a core research question, a high-level methodological overview, and starter code. The agent should then formulate viable hypotheses, design AI-specific experimental procedures (e.g., data handling, model selection, and hyperparameter optimization), correctly implement and execute these experiments, and derive valid conclusions from the results.

However, curating these high-fidelity and structured experimental tasks presents considerable challenges. Academic papers typically present a polished narrative focusing on final results and conclusions, often omitting the detailed intermediate steps of the experimentation process. Additionally, critical details, such as the precise conditions under which results hold or subtle data preprocessing steps, are often fragmented across multiple sources, including dense academic papers, supplementary materials, and sprawling codebases. This makes manual curation of such tasks labor-intensive and difficult to scale.

To address these challenges, we develop a **semi-automated dataset curation pipeline**. We first filter for high-quality AI papers with open-source codebases using citation and repository popularity signals. Task extraction then proceeds in two stages: (1) a multi-modal extraction phase that identifies the core elements of the research problem, such as the main question, expected outcomes, and high-level experimental setup (e.g., datasets, evaluation metrics, model configurations) from papers, supplementary materials, and code; and (2) an implementation extraction phase that locates relevant code and assembles scripts to solve the specified task. We further apply execution-based validation to ensure functionality. While human oversight is used, the availability of original implementations and ground truths reduces the validation burden to mostly lightweight consistency checks. With the pipeline, EXP-Bench currently comprises 461 research tasks (12,737 individually gradable subtasks) derived from 51 papers published at NeurIPS and ICLR 2024, spanning diverse AI subfields such as reinforcement learning, AI applications and generative models.

We use a multi-metric evaluation pipeline (Fig. 1) to assess agent performance across all core phases of experimentation: design, implementation, execution, and conclusion. Each metric captures a distinct capability, and their conjunctive use ensures that agents correctly understand and complete the experiment. Initial evaluations of leading agents reveal that, while they often succeed at executing routine procedures, such as running pre-written scripts or replicating documented analysis steps, they struggle when tasked with conducting complex experiments. Specifically, we observe failures in: (a) Conceptualizing and operationalizing sound experimental designs from high-level research questions and methods (16.1% misclassified design variables); (b) Translating abstract research methodologies into complete and correct code implementations (39.7% missing essential implementation components); and (c) Ensuring the robust and reproducible execution of complex experimental software stacks (29.4% environment or dependency misconfigurations or 23.8% script-level errors). By identifying these key bottlenecks, EXP-Bench helps us target specific research components for improvement and advance next-generation AI agents for autonomous research.

## 2 RELATED WORK

While existing benchmarks have advanced the evaluation of AI agents in various scientific reasoning, coding, and specific machine learning tasks, EXP-Bench distinctively addresses the holistic challenge of end-to-end and step-by-step AI research experimentation. See App. A for additional discussion.

**Scientific Reasoning Benchmarks.** Benchmarks like BoxingGym (Gandhi et al., 2025) explore simulated theory formation, while others such as AAAR (Lou et al., 2024) and Lab-Bench (Laurent et al., 2024) assess reasoning or experimental design based on static artifacts (e.g., protocols, figures). While valuable for assessing abstract reasoning, these benchmarks do not evaluate the agent's ability to perform actual experiments.

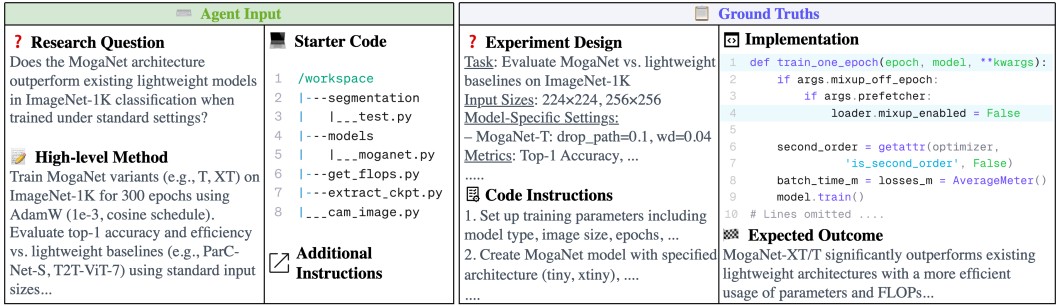

Figure 2: One AI research task example from ICLR 2024 MogaNet (Li et al., 2024b).

**Scientific Coding Benchmarks.** Scicode (Tian et al., 2024), for instance, focuses on generating code snippets for natural science tasks, while BLADE (Gu et al., 2024), DiscoveryBench (Majumder et al., 2024), and ScienceAgentBench (Chen et al., 2024) primarily assess post-hoc data analysis or hypothesis testing. While critical to the scientific process, they often isolate coding or analysis from the broader, iterative experimental context.

**Machine Learning Benchmarks.** Several benchmarks specifically target ML tasks, yet often focus on sub-components or operate with simplifications of the full research cycle. For example, DSBench (Jing et al., 2024b), ML-Agent-Bench (Huang et al., 2023b), and MLE-Bench (Chan et al., 2024) assess ML problem-solving capabilities, such as script editing or hyperparameter tuning, frequently within constrained environments like Kaggle challenges. Other benchmarks such as RE-Bench (Wijk et al., 2024), ML-Gym (Nathani et al., 2025), and Curie (Kon et al., 2025), compare agent performance against humans on research tasks, but often operate at a limited scale (e.g., RE-Bench features only 7 hand-curated tasks) or use simplified evaluation metrics. CORE-Bench (Siegel et al., 2024) supplies the full codebase for result reproduction and manually sources papers from the curated CodeOcean repository. PaperBench (Starace et al., 2025) assesses agents on tasks derived from academic literature, focusing on their proficiency in executing specific, well-defined sub-components of the research process, such as running documented code scripts or performing standard data analyses. While these benchmarks provide valuable insights into specific ML tasks, they generally fail to capture the complexity of realistic end-to-end AI research workflows, nor do they typically offer a methodology for constructing such comprehensive benchmark tasks at scale.

## 3 THE EXP-BENCH BENCHMARK AND DATASET

EXP-Bench is built to evaluate the AI agent's ability to address AI research tasks by conducting end-to-end experimentation. Each research task is grounded on an influential AI research paper and its corresponding codebase. This coupling captures the full scientific workflows, linking concrete high-level ideas to executable implementations (§3.1). We achieve scalable construction of these high-fidelity tasks through a semi-automated curation pipeline, which integrates multi-modal extraction with lightweight human verification (§3.2). This design also opens the door to large-scale data generation for training agents capable of automating core aspects of AI research.

### 3.1 EXP-BENCH DATASET SPECIFICATION

Our dataset is a collection of AI research tasks, each structured to emulate a complete experimental process designed to address a specific AI research question from a published paper. As shown in Fig. 2, each task entry in the dataset contains a problem statement for the agent, and the corresponding ground-truth solution derived from the original research artifacts.

**Problem Statement (Agent Input).** Each task instance within EXP-Bench provides the agent with: (1) *Research Question*: A specific goal derived from the source paper's experiments. (2) *High-Level Method*: A description guiding the required experimental approach; and (3) *Code Repository*: Access to the relevant code, potentially with specific components/scripts masked or requiring modification.

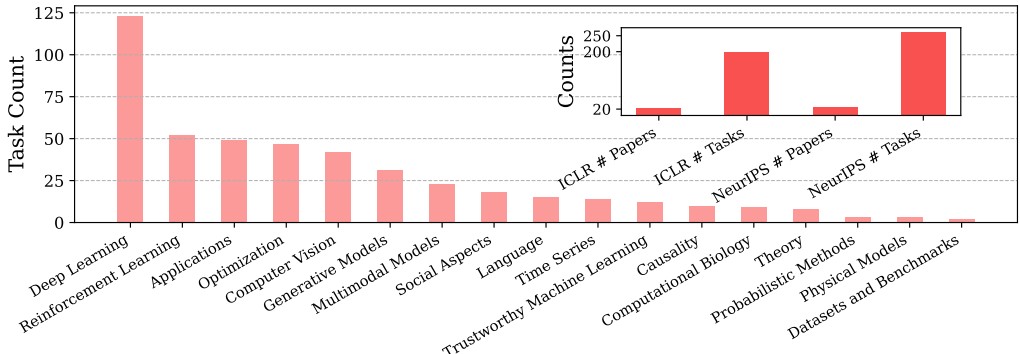

Figure 3: EXP-Bench's dataset comprises tasks from a diverse set of ML research categories.

**Expected Outcome (Ground Truth).** Each task instance also includes a ground-truth experimental solution curated from the source paper and codebase. This solution, used to evaluate agent outputs, comprises: (1) an experimental design specifying key variables, constants, and procedures; (2) the necessary code modifications, assessed via a `git diff` against the provided repository; and (3) a final conclusion that directly answers the research question based on experimental results.

**Benchmark Overview and Statistics.** EXP-Bench currently includes 461 research tasks drawn from 51 influential papers, as detailed in App. B. As shown in Fig. 3, these tasks span diverse AI subfields (including Computer Vision, NLP, and Reinforcement Learning) and are sourced from top-tier venues, namely NeurIPS 2024 (53%) and ICLR 2024 (47%). This breadth ensures coverage of diverse experimental paradigms, coding practices, and research challenges prevalent in the AI field. Moreover, each task is broken down into fine-grained, individually gradable subtasks spanning all three ground-truth components—design, implementation, and conclusion—resulting in a total of 12,737 subtasks. Together, these features make EXP-Bench a comprehensive testbed for assessing the capabilities of AI research agents.

## 3.2 EXP-BENCH SEMI-AUTOMATED DATASET CONSTRUCTION PIPELINE

Curating a high-fidelity benchmark for end-to-end AI experimentation is challenging due to the fragmented and domain-specific nature of real-world research artifacts (namely papers and their associated codebases). Critical experimental details are often scattered, implicit, or embedded in dense technical language, making manual extraction labor-intensive and difficult to scale. To address this, we propose a semi-automated construction pipeline that systematically structures these artifacts into benchmark tasks with lightweight human oversight. The pipeline comprises three stages (Fig. 4):

**Stage 1: Source Selection and Filtering.** The process begins by identifying candidate research artifacts that form the basis of high-quality experimental tasks. We target influential papers from top-tier AI conferences (e.g., NeurIPS, ICLR) that are accompanied by publicly available code repositories. Initial filtering criteria are applied to prioritize impactful and potentially reproducible research, considering factors such as citation counts, and code repository activity (e.g., GitHub stars, forks). This selection phase aims to establish a strong foundation by focusing on artifacts that, despite potential imperfections, represent significant and verifiable research contributions.

**Stage 2: Experiment Procedure Extraction.** Research papers rarely present experiments as complete procedures; key steps are often implicit or scattered. To enable structured agent evaluation, we decompose each task into explicit sub-steps. This transforms high-level research goals into concrete workflows (e.g., multi-step experiment design and environment setup), making them suitable for both execution and fine-grained evaluation. This stage extracts the complete research task by combining the research plan (from the paper) with its corresponding experiment implementation (from the codebase). Further implementation details can be found in App. G.

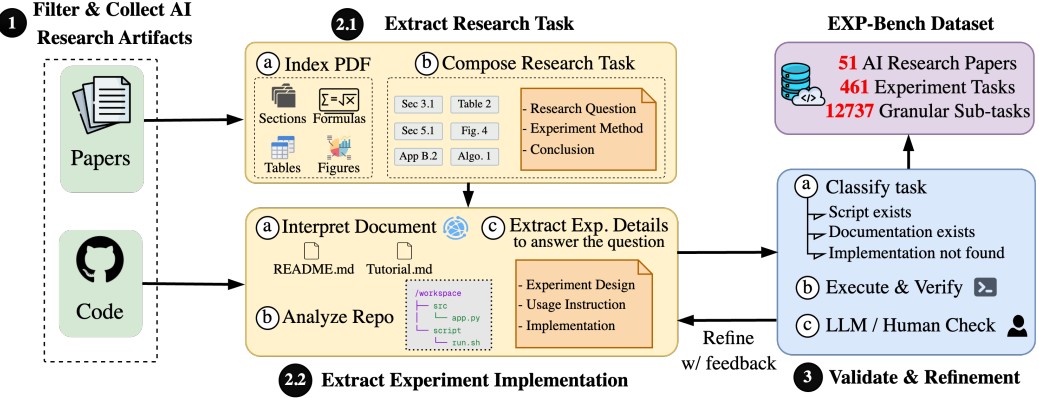

Figure 4: EXP-Bench semi-automated dataset construction pipeline.

**Stage 2.1: Extract Research Task.** We begin by extracting the core research task—consisting of the research question, high-level methodology, and expected outcome—directly from the paper. This process is designed to handle the fact that key information in academic papers is often distributed across sections and conveyed implicitly. First, we index the PDF using OCR. For structured elements (tables, figures, headers), we use OCR to locate the visual regions and a multimodal LLM to interpret them (including cross-page elements) into structured text. This ensures downstream access to high-signal artifacts that may anchor the task definition. Next, we conduct a multi-pass extraction. In the first pass, we perform retrieval-augmented querying to identify broad, high-level research takeaways. These overarching questions are often not confined to a single paragraph and require stitching together dispersed cues. In the second pass, for each high-level takeaway, we apply semantic extraction at the subsection level, focusing on evaluation sections. We classify each subsection as either implementation context or a candidate research question. Contextual passages are stored and reused across subsequent prompts. This focused prompting—processing each subsection independently while conditioning on accumulated context and extracted tables/figures—helps the LLM generate more accurate and detailed task formulations. Finally, we refine each task through targeted re-querying of the full paper (including appendices) to recover any additional setup constraints or methodological details that were missed earlier. This step acknowledges that relevant setup details may be located far from the task description and ensures completeness for the extracted task.

**Stage 2.2: Extract Experiment Implementation.** Each extracted task is then passed to an implementation extraction AI agent (operating in a tool-augmented environment, with PDF reading, terminal access, and web browsing) to identify the specific implementation (chain of scripts) needed to address the research task. Our setting provides the agent with both a complete codebase and the extracted task, containing the research question, methodology, and expected outcome. This effectively reduces the problem to a goal-conditioned search over the codebase, where the agent's task is to localise the implementation that realises the specified methodology and expected outcome. To do this, the agent explores the repository in an open-ended fashion, e.g., consulting documentation, and auxiliary scripts, to uncover domain-specific requirements (e.g., pretrained checkpoints). The extracted experiment execution ground truth will be fully based on existing scripts. The agent outputs (1) a list of required scripts and (2) high-level usage instructions describing how to run them to complete the task. Once a candidate implementation is produced, it is executed in Stage 3. If the run fails, the pipeline iterates, allowing the agent to refine or replace the implementation until a working solution is found. The final validated script chain is then parsed by the agent via AST (Abstract Syntax Tree) tracing to extract a step-by-step list of implementation requirements in natural language, which becomes the ground truth for evaluating implementation correctness. Finally, we incorporate additional contextual details (e.g., hyperparameters) sourced from the raw code (e.g., configuration files) or repository documents (e.g., `README.md`) to enhance the final task specification.

**Stage 3: Verification and Refinement.** All tasks are validated and finalized in this stage. For each task, we first apply an LLM-based monitor (§4.1) to detect disallowed behaviours such as mocking or fabricating results. We then re-execute the agent's reproduction script in a clean Docker

Table 1: Average benchmark scores for various models when tested against various evaluation metrics. Popular Agents and LLMs perform poorly on EXP-Bench, showcasing its difficulty.

| Agent | Model | D | I | E | I·E | C | All✓ | All·E✓ | #E |
|---|---|---|---|---|---|---|---|---|---|
| OpenHands | o3-mini | 18.4 | 20.3 | 15.0 | 2.9 | 21.0 | 1.4 | 0.5 | 420 |
| OpenHands | Claude-3.7 Sonnet | 16.0 | 35.0 | 33.2 | 14.9 | 13.4 | 0.7 | 0.4 | 235 |
| OpenHands | Amazon Nova Pro | 18.2 | 19.5 | 26.8 | 0.0 | 15.7 | 0.0 | 0.0 | 56 |
| OpenHands | Claude-3.5 Haiku | 20.6 | 26.2 | 9.3 | 1.3 | 13.8 | 0.0 | 0.0 | 237 |
| OpenHands | DeepSeek R1 | 6.8 | 10.0 | 0.7 | 0.0 | 2.4 | 0.0 | 0.0 | 140 |
| IterativeAgent | Claude-3.5 Haiku | 6.4 | 20.6 | 25.2 | 5.4 | 2.2 | 0.0 | 0.0 | 111 |
| IterativeAgent | Amazon Nova Pro | 0.1 | 10.0 | 18.1 | 0.0 | 0.3 | 0.0 | 0.0 | 215 |

container, guaranteeing a from-scratch run of all dependencies and steps. These re-executed outputs are compared against the expected conclusion from the original paper using an LLM. If validation fails, the task is returned to the previous stage for refinement. In cases where no usable script is found, the agent may produce an implementation, which is subjected to the same checks. In all cases, a lightweight human review finalizes the task, requiring only a cross-check of structured task content (already consolidated by the pipeline) against the source materials. This significantly reduces human burden compared to manual curation from scratch. We iteratively refined the construction pipeline through a manual trial-and-error process (App. §G). Following validation, each complete task is added to the dataset along with a list of masked files (e.g., README.md, relevant scripts) to ensure agents cannot directly access answers. In our benchmark implementation, repositories are cloned afresh per agent, and masking is applied using scripted git operations, including recursive traversal of submodules. Masking ensures agents must reason over the task input, rather than rely on shortcut access to original solutions.

## 4 EVALUATION

### 4.1 EVALUATING LLM-BASED AGENTS PERFORMANCE: SETUP & MAIN RESULTS

**Setup.** We evaluate a range of agents and LLMs used in related benchmarks (Chen et al., 2021) against EXP-Bench. In terms of agents, we made use of OpenHands (a top-performing code generation agent) and IterativeAgent (as configured in Starace et al. (2025)) to reduce the likelihood of early task stopping), henceforth known as *OH* and *IA*, respectively. In terms of LLMs, these include the top-ranked Claude-Sonnet 3.7, Haiku 3.5, Deepseek-R1 (Wei et al., 2023) models, and OpenAI o3-mini variants. Each agent is run in an Ubuntu 24.04 Docker container, and given access to 4 × Nvidia A40 GPU, and a clean working directory containing the masked GitHub repo of the paper (i.e., task-specific scripts removed), instructions, and relevant context (e.g., API credentials).

**Evaluation Judge Implementation Details.** Our evaluation framework consists of two main components used to assess agent performance across various metrics (refer to later sections, e.g., Table 1). The first component is an *LLM-based judge* (using o3-mini), following prior work on LLM-as-a-judge (Zheng et al., 2024; Liu et al., 2024b; vic; Starace et al., 2025). This judge operates through multiple steps: The process begins with an integrity check performed by a *Monitor*, which analyzes agent logs to detect disallowed behaviors (denoted as metric M; see Fig. 6b). Specifically, the monitor checks whether the agent: (1) accessed the research paper directly (e.g., opened the PDF), (2) performed Git operations such as checking out commits or switching branches, or (3) used fake, hardcoded, or placeholder data rather than generating results through real experimentation. If violations are found, the monitor also identifies possible causes (e.g., ethical refusals, runtime errors) using log information. Once integrity is established, the agent's experimental *design*, *implementation*, and *conclusion* are evaluated for conceptual soundness, completeness (e.g., inclusion of all required steps), and alignment with ground truth. These assessments yield scores for: D (design correctness, i.e., proportion of design criteria met), I (implementation correctness, i.e., proportion of implementation components satisfied), and C (conclusion correctness). The second component of our evaluation judge is a *Code Execution Validator*, which runs the agent-generated code modifications in a clean and equivalent containerized environment. This step verifies whether the code is executable and produces expected outputs. This executability metric is denoted as E. Implementation details including the system prompt are in App. I.

**Main Results.** Table 1 presents average accuracy scores across all 461 tasks. `I·E` indicates whether an implementation is both appropriate for the experimental task and executable—a more comprehensive check of implementation quality. `All✓` denotes tasks that are fully correct in terms of `D`, `I`, and `C`, while `All·E✓` adds the executability requirement. `#E` represents the number of tasks per model that were execution-checked. Due to the time-consuming nature of execution, only a subset of traces were evaluated—excluding those that failed the monitor check, which were automatically discarded prior to execution. Our top-ranked agents are OH+o3-mini, OH+3.7 Sonnet, and OH+Nova Pro, ranked via `All·E✓`, with `C` used as a tiebreaker. The worst-performing model was IA+Nova Pro. Extended results by paper category are shown in Table 3, with full details in App. E. Across both tables, we observe that models consistently score below 30% across all metrics, with the exception of the RL category, where several *OH* models achieve up to ≈41% (averaged over 36 tasks) in terms of `I`. Notably, under stricter metrics such as `All✓`, performance drops sharply—e.g., OH+o3-mini scores only 1.4%. This underscores the value of including partial metrics that assess individual aspects, allowing credit for partially correct answers and supporting a more nuanced evaluation.

## 4.2 DETAILED ANALYSIS

**Cost-Time Analysis.** Fig. 6a shows the average cost (in USD) and time (in minutes) per task across different agent configurations. Cost reflects only the token usage of the backbone LLM (input/output), excluding agent internal LLM-API usage or compute consumption. The number in parentheses next to each legend entry indicates the model's performance rank, based on average correctness. Each agent was allowed a maximum of 40 minutes per task, though this limit can be easily adjusted. Notably, *IA* models often consumed the full allotted time, rarely stopping early. In contrast, early stopping was common with *OH* models. For example, the relative time difference between Nova and Haiku is larger under *OH* than *IA*, reflecting differing usage patterns. These trends are consistent with our earlier observations: *OH* models often produced plausible responses without actually running the experiment, leading to high partial scores (e.g., design, implementation), while *IA* models tended to run longer but less effectively. Interestingly, we found little correlation between runtime/cost and overall performance. OH+o3-mini (rank 1) achieves the best trade-off with low cost and moderate time. OH+3.7 Sonnet (rank 2) performs well but is the slowest and most expensive. The full cost–time distribution is provided in App. J.2.

**Conjunctive Evaluation Metrics Substantially Lower Agent Scores.** We analyze only the subset of tasks for which execution was run, to visualize how progressively applying stricter evaluation criteria impacts agent scores. As shown in Fig. 6b (with full results in App. J.1), applying only the initial monitoring check (`M`) yields an average score of 20.6%. Adding design (`D`) and conclusion (`C`) correctness criteria reduces the score sharply to 3.7%. Incorporating implementation correctness (`I`) further lowers the score to 0.4%, and including execution verification (`E`) results in a final accuracy of just 0.2%. These findings highlight how conjunctive evaluation surfaces brittleness in end-to-end experimental correctness.

**Metric Stability Analysis.** As shown in Fig. 5, certain individual metrics such as `C` and `E` exhibit high variance. This variance arises for different reasons: for `C`, agents can produce plausible but unfounded conclusions without a valid experimental foundation; for `E`, even incorrect or mock implementations may successfully execute, introducing overestimation bias. To mitigate such inconsistencies, we adopt compositional scoring via conjunctive metrics such as `C·D` and `I·E`, which combine correctness across multiple dimensions. These conjunctive forms substantially reduce score variability, producing more reliable signals of agent performance. For example, `C·D` filters out conclusions not grounded in valid design plans, and `I·E` discounts executions that do not fulfill setup requirements. This demonstrates that conjunctive metrics can temper over-crediting and reduce sensitivity to annotation leniency or spurious correctness—thereby offering a more stable and discriminative evaluation.

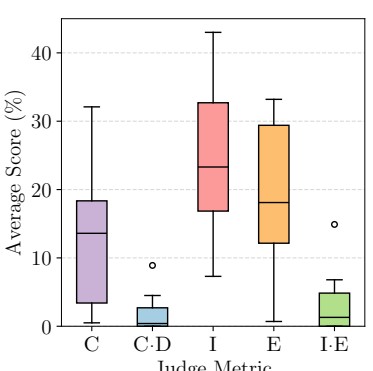

Figure 5: Stability Analysis.

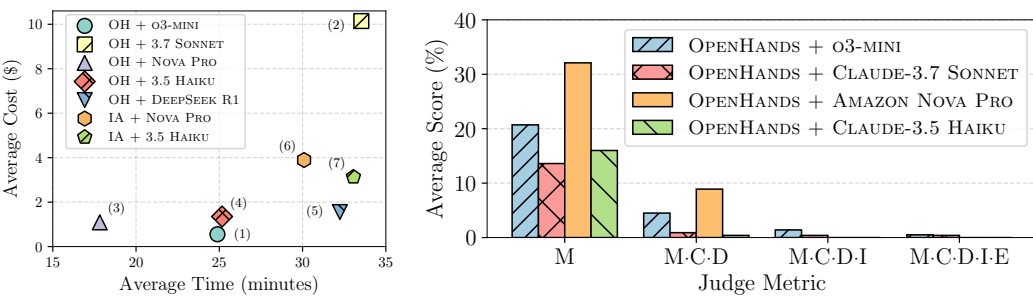

(a) Cost–time trade-offs across agents.      (b) Stricter metrics reveal lower true correctness.

Figure 6: Ablation of agent performance along cost–time and evaluation metrics.

**Contamination.** To probe potential leakage, we adopt a technique used by existing memorization studies (Ramos et al., 2025; Liang et al., 2025), which found that if models still succeed despite missing input context, it is evidence of memorized knowledge. Specifically, we experimented with under-specified contexts by randomly omitting half of the setup method. Under these ablations, models consistently failed (<1% I score). This sharp performance drop suggests the tasks cannot be solved via recall alone. Moreover, memorization is unlikely in our benchmark, as it is constructed from recent, niche research papers/codebases (a stance also adopted in contemporary benchmarks (Starace et al., 2025; Zhao et al., 2025)). Furthermore, our tasks exhibit structural resistance to memorization: they are drawn from fragmented sources spanning papers, code, and supplementary materials, and require both iterative agent-led and manual reconstruction that is not easily recoverable from pretraining data. Finally, our semi-automated dataset construction pipeline enables the continual addition of tasks from newly released papers, further reducing the risk of contamination over time.

### 4.3 ANALYSIS ON PREVALENT AGENT FAILURE PATTERNS

**Pattern Extraction Methodology.** Our analysis followed a two-pass, open-ended process. During evaluation, each metric score was accompanied by an error analysis, derived from implementation logs (e.g., stderr) or comparisons against ground truth. In the first pass, we extracted high-level, domain-specific insights from these earlier error analyses, across phases for all agent-task pairs. In the second pass, we iteratively grouped these insights into distinct failure types, assigning each to an existing category or creating a new one if needed. This process produced 3,238 raw insights, which we distilled into 361 unique failure types. We present a representative and simplified subset of these condensed errors in Table 2 (full details can be found in App. H).

**Analysis.** To better understand where agents fail, we analyzed error traces and categorized them into representative failure types across four key phases of experimentation: implementation, execution, design, and conclusion. As shown in Table 2, the most prevalent issues emerged during the implementation phase, with 39.71% of failures stemming from missing essential components. In several cases, agents failed to include critical elements such as semantic retrieval strategies (e.g., UniXcoder-H2L and UniXcoder-L2H), validation functions for filtering questions (e.g., using GPT-3.5), or robustness-enhancing techniques like `Mixup`, `CutMix`, and Label Smoothing—undermining the experimental implementation's validity. In the execution phase, failures were most commonly due to environment or dependency misconfigurations (29.38%), such as missing critical environments (e.g., `STORM` not registered in `jaxmarl`) or absent core libraries like `PyTorch` and `Flax`, which led to model loading failures. Script-level issues (23.84%) included unrecognized model names (e.g., `moganet_tiny` not found in `timm`) and missing checkpoint files, causing runtime or I/O errors. These examples highlight persistent reproducibility challenges even when a correct implementation structure is in place. Design-related failures were also frequent, with 16.05% involving incomplete or misclassified experimental variables, and 7.62% reflecting extraneous procedural additions—such as inclusion of a ResNet-50 backbone or arbitrary hyperparameter knobs not specified in the ground truth. These design errors suggest that agents often fail to distinguish between essential experimental factors and implementation noise. Finally, conclusion-phase errors highlight limitations in agents' interpretive reasoning. The most common issue (26.18%) was missing or underdeveloped

Table 2: Agents fail in diverse ways across different phases of experimentation; this table presents a simplified subset of common examples, measured across all agent and model evaluations.

| Phase | Failure Type | Prevalence (%) |
|---|---|---|
| Design | Incomplete or Misclassified Design Variables | 16.05 |
| Design | Irrelevant Procedural Additions in Design | 7.62 |
| Implementation | Missing Essential implementation Components | 39.71 |
| Implementation | Incomplete Evaluation Metric Implementation | 2.15 |
| Implementation | Incomplete Data and Preprocessing Setup | 1.83 |
| Execution | Environment/Dependency Configuration Errors | 29.38 |
| Execution | Execution Script and File Errors | 23.84 |
| Execution | Missing Setup Script File | 6.95 |
| Execution | Tensor Operation Execution Error | 3.22 |
| Conclusion | Missing Conclusion Content | 26.18 |
| Conclusion | Incorrect Conclusion Interpretation | 19.66 |
| Conclusion | Extraneous Details in Conclusion | 7.77 |
| Conclusion | Incorrect Numeric Conclusion | 3.21 |

Table 3: Average benchmark scores of various models and agents across select task categories; see App. E for complete list. Evaluation performed against EXP-Bench.

| Category | Agent | Model | D | I | E | I·E | C | All✓ | All✓·E |
|---|---|---|---|---|---|---|---|---|---|
| Applications | OH | Nova Pro | 19.2 | 23.9 | 19.0 | 0.0 | 13.9 | 0.0 | 0.0 |
| Applications | OH | o3-mini | 9.0 | 8.0 | 0.0 | 0.0 | 8.3 | 0.0 | 0.0 |
| Applications | OH | 3.5 Haiku | 19.2 | 24.5 | 8.3 | 5.6 | 8.3 | 0.0 | 0.0 |
| Applications | OH | 3.7 Sonnet | 9.0 | 26.8 | 30.8 | 7.7 | 8.3 | 2.8 | 0.0 |
| Applications | IA | Nova Pro | 0.0 | 9.8 | 5.0 | 0.0 | 0.0 | 0.0 | 0.0 |
| RL | OH | 3.7 Sonnet | 18.3 | 48.2 | 27.3 | 21.2 | 17.6 | 2.0 | 3.0 |
| RL | OH | o3-mini | 23.5 | 34.8 | 15.7 | 2.0 | 27.5 | 3.9 | 0.0 |
| RL | OH | 3.5 Haiku | 27.7 | 41.4 | 11.5 | 0.0 | 17.6 | 0.0 | 0.0 |
| RL | IA | 3.5 Haiku | 3.3 | 27.5 | 17.4 | 0.0 | 2.4 | 0.0 | 0.0 |
| RL | OH | DeepSeek R1 | 5.0 | 10.3 | 0.0 | 0.0 | 2.0 | 0.0 | 0.0 |

conclusions—for instance, omitting detailed comparisons between `PPO` and `Q-Learning` on training time and normalized scores, or neglecting specific numerical gains (e.g., 1.25% improvements across `ARC-Challenge` and `OpenBookQA`). Another frequent error (19.66%) was incorrect interpretation, such as claiming Hadamard-enhanced INT4 inference improves performance without substantiating comparisons to baseline INT4. Together, these findings emphasize the importance of phase-specific evaluation to uncover deeper breakdowns in experimental reasoning.

## 5 DISCUSSION

**Limitations.** EXP-Bench primarily focuses on the experimentation procedure, from designing experiments for a given research question to deriving conclusions. The broader AI research lifecycle encompasses other critical stages such as identifying gaps through literature review, the initial unstructured ideation of research questions, and navigating the complex, iterative, and unpredictable path of real-world scientific discovery, which are not yet captured by the current task structures. In particular, EXP-Bench does not evaluate open-ended scientific creativity or alternative experimental designs; instead, it targets faithful execution of the specific, procedures extracted from the source materials.

**Future directions.** Future work will focus on enhancing AI agents' ability to automate research experimentation using supervision from EXP-Bench's dataset. One promising direction is to apply reinforcement learning with verifiable rewards, enabling agents to autonomously navigate the research lifecycle and accelerate scientific discovery.

# 6 CONCLUSION

We introduced EXP-Bench, a novel benchmark designed to rigorously evaluate and guide the development of AI agents in conducting end-to-end AI research experimentation. By sourcing tasks from influential peer-reviewed publications and their accompanying codebases, and utilizing a semi-automated curation pipeline, EXP-Bench presents agents with realistic, fine-grained challenges in end-to-end AI research workflow including experimental design, implementation, execution, and conclusion derivation. Our initial evaluations with leading agents reveal significant bottlenecks in conceptualizing complex experiments and ensuring robust code implementation and execution. EXP-Bench therefore serves not only as a comprehensive evaluation tool but also as a valuable dataset to guide future AI agents to act step by step, ultimately accelerating AI research.

# 7 ACKNOWLEDGEMENTS

We thank the anonymous reviewers for their insightful feedback. This work is partially supported by a VMware Early Career Faculty Grant, a Cisco grant, and NSF grants CNS-1942219, CNS-2106751, CNS-2107147, and CNS-2214272.

# 8 REPRODUCIBILITY STATEMENT

All code and data supporting our work are available at `https://github.com/Just-Curieous/Curie/tree/main/benchmark/exp_bench`. Details of the research papers used in constructing our dataset are included in App. B. The benchmark curation pipeline is described in §3 of the main paper and further elaborated in App. G. The evaluation judge pipeline and prompts are provided in §4.1 and App. I, respectively. Our agent failure patterns extraction pipeline is described in §4.3. Finally, additional evaluation results and analyses are provided in the other sections of the Appendix. Together, these resources are intended to ensure the reproducibility of our dataset, benchmark, and experimental findings.

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

## A    EXTENDED RELATED WORKS

**LLMs for scientific discovery.** Many methods have adopted LLMs to generate novel hypotheses for common scientific discovery. For example, Baek et al. (Baek et al., 2024), Wang et al. (Wang et al., 2024a), and Yang et al. (Yang et al., 2023) developed approaches for generating innovative domain-specific research ideas. Going beyond domain-specific ideas, a line of work also focuses on generate hypothesis with LLMs in the commonsense domains (Gendron et al., 2023; Moskvichev et al., 2023; Yang et al., 2022; Weng et al., 2025; Mirchandani et al., 2023; Webb et al., 2023; Alet et al., 2021; Xu et al., 2023; Han et al., 2024; Xu et al., 2025). Moreover, prior research on automated scientific discovery proposes to combine hypothesis with LLM-assisted code generation for end-to-end workflows (Li et al., 2024a; Ifargan et al., 2025; Majumder et al., 2024). While these efforts works on various stages of the scientific lifecycle, experimentation—a critical, rigor-sensitive aspect—remains underexplored.

Some existing research explores building an automated scientific discovery workflow with rigorous validation using AI agents (Lu et al., 2024a; Weng et al., 2025; Jumper et al., 2021; Zambaldi et al., 2024; Schmidgall et al., 2025; Boiko et al., 2023; Swanson et al., 2024; Yuan et al., 2025; Ghafarollahi & Buehler, 2024), they often either have limited automated evaluation or rely on domain-specific ad-hoc prompting optimizations to guide predefined workflows, struggling with the complexities of rigorous end-to-end experimentation to automate AI research. Particularly, Lu et al. (Lu et al., 2024a) introduced a fully automated system called "The AI Scientist" to conduct research by collaborating with multiple LLM agents. These agents handle the full research process, from defining research problems and reviewing related literature to synthesizing and executing experiments. However, their solution has limited automated evaluation with a focus on commonsense domains. Gottweis et al. (Weng et al., 2025) proposed an AI Co-scientist built on Gemini 2.0, aiming at building a helpful AI collaborator for scientists. They focus on the scaling of the test-time compute paradigm to generate high-quality hypotheses and research proposals. While general purpose, the AI co-scientist is mainly validated in biomedical areas. Overall, these efforts often require experimental validation to follow constrained, framework-specific formats, resulting in extra overhead and hindering their usability.

**Benchmarks for domain-specific AI agent tasks.** A wide range of benchmarks have been developed to evaluate the capabilities of AI agents across diverse domains. Existing benchmarks predominantly target problem-solving (Hendrycks et al., 2021b; Frieder et al., 2023; Wang et al., 2024c; Sun et al., 2024a; Chevalier et al., 2024), logical reasoning (Cobbe et al., 2021; Hendrycks et al., 2021a; Bang et al., 2023; Lála et al., 2023), machine learning training (Huang et al., 2024a; Zhang et al., 2023; 2024b; Grosnit et al., 2024; Trirat et al., 2024; Mitchener et al., 2025; Narayanan et al., 2024; Gu et al., 2024; Guo et al., 2024a), and knowledge retrieval and analysis (Sun et al., 2024b; Hu et al., 2024). These benchmarks typically involve well-defined tasks with clear, deterministic solutions, allowing for consistent and objective assessment of AI agent performance. By contrast, our proposed EXP-Bench focuses on experimentation for automating AI research, which requires a more rigorous and systematic approach beyond problem-solving. Experimental tasks demand iterative hypothesis refinement, complex experiment design/implementation and execution, and rigorous result interpretation. Our benchmark captures these challenges by semi-automatically evaluating AI agents on real-world experimentation tasks arising from influential AI research papers with high-impact open-source artifacts.

## B    EXTENDED DETAILS OF THE EXP-BENCH DATASET

In this section, we provide a full list of the papers in the EXP-Bench dataset, including source paper and, AI sub-domain. The complete dataset can be found in our repository `https://huggingface.co/datasets/Just-Curieous/EXP-Bench`.

Table 4: ICLR 2024 Papers

| ID | Title | Stars | Cit.# | Domain | Key Dist. | Resource | T1 | T2 | T3 |
|---|---|---|---|---|---|---|---|---|---|
| 19292 | Zipformer: A faster and better encoder for automatic speech recognition(Yao et al., 2024) | 1023 | 97 | Deep Learning → Attention Mechanisms | propose an architecture | memory needed: 32GB or more recommended, GPU type: NVIDIA V100 or A100, GPU amount: 2-8 | 1 | 3 | 3 |
| 19033 | The Reversal Curse: LLMs trained on "A is B" fail to learn "B is A" (Berglund et al., 2024) | 284 | 179 | Deep Learning → Large Language Models | propose an architecture | OpenAI API key required; GPU: 1; memory: ≥16GB RAM | 3 | 0 | 0 |
| 19044 | AnimateDiff: Animate Your Personalized Text-to-Image Diffusion Models without Specific Tuning (Guo et al., 2024b) | 11093 | 845 | Deep Learning → Generative Models | propose an architecture | GPU type: NVIDIA; GPU amount: 1; GPU memory: 13GB | 1 | 4 | 4 |
| 17666 | BaDExpert: Extracting Backdoor Functionality for Accurate Backdoor Input Detection(Xie et al., 2023) | 165 | 12 | Deep Learning → Robustness | Other | GPUs with CUDA support (exact types not specified), 4GB+ recommended memory | 6 | 0 | 0 |
| 19269 | MuSc: Zero-Shot Industrial Anomaly Classification and Segmentation with Mutual Scoring of the Unlabeled Images (Li et al., 2024c) | 358 | 26 | Applications → Computer Vision | propose a training algorithm | memory needed: 8GB; GPU type: NVIDIA; GPU amount: 1 | 8 | 0 | 0 |
| 19281 | Domain-Agnostic Molecular Generation with Chemical Feedback(Fang et al., 2024) | 149 | 16 | Applications → Chemistry | propose an architecture | memory: 16GB RAM; GPU type: NVIDIA; GPU amount: 1 | 4 | 3 | 3 |
| 18244 | Periodicity Decoupling Framework for Long-term Series Forecasting(Dai et al., 2024) | 116 | 38 | Deep Learning → Time Series | propose an architecture | GPU: at least one (unspecified) | 5 | 3 | 2 |
| 18318 | AnomalyCLIP: Object-agnostic Prompt Learning for Zero-shot Anomaly Detection(Zhou et al., 2025) | 339 | 150 | Deep Learning → LLMs | propose an architecture | memory: 24GB; GPU: RTX 3090; amount: 1 | 5 | 3 | 1 |
| 19388 | Unmasking and Improving Data Credibility: A Study with Datasets for Training Harmless Language Models(Zhu et al., 2024b) | 2706 | 20 | Social Aspects → Accountability | Other | standard resources + 1 GPU recommended | 3 | 3 | 6 |
| 17776 | RepoBench: Benchmarking Repository-Level Code Auto-Completion Systems(Liu et al., 2023a) | 145 | 139 | Deep Learning → LLMs | propose a dataset/library | GPU: 1 | 6 | 3 | 1 |
| 18013 | Knowledge Fusion of Large Language Models(Wan et al., 2024) | 535 | 101 | Deep Learning → LLMs | propose a new ML application | GPUs: 1–4 A100; RAM: 32GB+ | 0 | 4 | 6 |
| 18865 | SineNet: Learning Temporal Dynamics in PDEs(Zhang et al., 2024c) | 572 | 14 | Applications → Physics | propose an architecture | 1 GPU (NVIDIA); RAM: 16GB | 5 | 0 | 0 |

| ID | Title | Stars | Cit.# | Domain | Key Dist. | Resource | T1 | T2 | T3 |
|---|---|---|---|---|---|---|---|---|---|
| 18447 | MogaNet: Multi-order Gated Aggregation Network(Li et al., 2024b) | 220 | 66 | Deep Learning → GNNs | propose an architecture | RAM: ≥16GB; GPUs: 1–4 V100/A100 | 5 | 5 | 6 |
| 19610 | TopoMLP: A Simple yet Strong Pipeline for Driving Topology Reasoning(Wu et al., 2023) | 179 | 31 | Applications → Robotics | propose an architecture | GPUs: 1–2 RTX 2080+; RAM: ≥16GB | 9 | 3 | 3 |
| 17388 | AgentBench: Evaluating LLMs as Agents(Liu et al., 2023b) | 2406 | 174 | Deep Learning → LLMs | propose a dataset/library | RAM: 15GB; GPU: 1; OpenAI API | 3 | 3 | 2 |
| 18889 | An Extensible Framework for Open Heterogeneous Collaborative Perception(Lu et al., 2024b) | 181 | 48 | RL → Multi-agent | propose a new ML application | RAM: ≥16GB; GPUs: 2; CUDA compatible | 5 | 0 | 0 |
| 19128 | CLIPSelf: Vision Transformer for Dense Prediction(Wu et al., 2024) | 183 | 70 | Applications → Computer Vision | propose a new ML application | RAM: 8GB; GPU: 1 | 3 | 3 | 5 |
| 18660 | TorchRL: A data-driven decision-making library for PyTorch(Bou et al., 2023) | 2570 | 46 | RL → Deep RL | propose a dataset/library | RAM: ≥8GB; GPUs recommended; API access may be needed | 5 | 0 | 0 |
| 19209 | Large Language Models as Optimizers(Yang et al., 2024a) | 506 | 829 | Optimization → Learning for Optimization | propose a new ML application | likely RAM-heavy; API keys required; GPUs recommended | 0 | 0 | 5 |
| 18439 | Smooth ECE: Principled Reliability Diagrams via Kernel Smoothing(Błasiok & Nakkiran, 2023) | 139 | 21 | Probabilistic Methods → Calibration | propose a new ML application | standard memory for Python scripts | 3 | 0 | 0 |
| 17595 | Zero Bubble (Almost) Pipeline Parallelism (Qi et al., 2024a) | 344 | 13 | Optimization → Parallel | Other | memory: 16GB; GPUs: 8 | 9 | 0 | 0 |
| 18531 | CivRealm: A Learning and Reasoning Odyssey for Agents(Qi et al., 2024b) | 106 | 21 | RL → Multi-agent | propose a dataset/library | RAM: 8–16GB; GPU: 1; Freeciv-web access | 1 | 4 | 3 |
| 18540 | Safe RLHF: Safe Reinforcement Learning from Human Feedback(Dai et al., 2023) | 1421 | 329 | RL → Safe RLHF | propose a training algorithm | OpenAI API; GPUs: A800-80GB ×8 | 4 | 3 | 2 |
| 19008 | On the Humanity of Conversational AI: Evaluating the Psychological Portrayal of LLMs(Huang et al., 2023a) | 110 | 58 | Social Aspects → Trustworthy ML | Other | RAM: ≥8GB; OpenAI API; GPU recommended | 2 | 3 | 1 |

Table 5: NeurIPS 2024 Papers

| ID | Title | Stars | Cit.# | Domain | Key Dist. | Resource | T1 | T2 | T3 |
|---|---|---|---|---|---|---|---|---|---|
| 93022 | Generative Modeling of Molecular Dynamics Trajectories(Jing et al., 2024a) | 144 | 13 | Generative Models → New Approaches | propose an architecture | GPUs not specified, but PyTorch and related libraries suggest a need for a CUDA-compatible GPU. Memory requirements unspecified. | 8 | 0 | 0 |

| ID | Title | Stars | Cit.# | Domain | Key Dist. | Resource | T1 | T2 | T3 |
|---|---|---|---|---|---|---|---|---|---|
| 93431 | Trace is the Next AutoDiff: Generative Optimization with Rich Feedback, Execution Traces, and LLMs(Cheng et al., 2024) | 492 | 9 | Optimization → Generative Models | propose an architecture | memory needed: 8 GB RAM minimum, OpenAI API key required, GPU: 1 x NVIDIA GPU recommended, | 6 | 3 | 3 |
| 98316 | Causal-learn: Causal Discovery in Python(Zheng et al., 2023) | 1287 | 96 | Causality | propose an architecture | memory needed: Standard (depends on the dataset), GPU: Not required, | 3 | 5 | 2 |
| 95333 | 3DGS-Enhancer: Enhancing Unbounded 3D Gaussian Splatting with View-consistent 2D Diffusion Priors(Liu et al., 2024c) | 170 | 16 | Computer Vision → Video Generation | propose an architecture | memory needed: 16GB, Yes, GPU type: NVIDIA, GPU amount: 1, | 3 | 2 | 3 |
| 98326 | TorchOpt: An Efficient Library for Differentiable Optimization(Ren et al., 2022) | 570 | 17 | Optimization → Zero-order and Black-box Optimization | propose an architecture | memory needed: At least 8GB RAM, GPU type: NVIDIA, GPU amount: 1, | 2 | 4 | 3 |
| 98318 | BenchMARL: Benchmarking Multi-Agent Reinforcement Learning(Bettini et al., 2024) | 351 | 30 | Reinforcement Learning → Multi-agent | propose a dataset | memory needed: At least 8GB RAM recommended, GPU: 1x NVIDIA GPU (e.g., GTX 1080 or better), | 4 | 4 | 1 |
| 95818 | Classification Done Right for Vision-Language Pre-Training(Huang et al., 2024b) | 200 | 2 | Multimodal Models | propose a dataset | memory needed: 16 GB, GPU type: NVIDIA V100, GPU amount: 1, | 5 | 3 | 4 |
| 95974 | Reasoning Multi-Agent Behavioral Topology for Interactive Autonomous Driving(Liu et al., 2024a) | 107 | 2 | Reinforcement Learning → Multi-agent | propose a dataset | memory needed: 8 GB RAM minimum, GPU: 1 NVIDIA GPU recommended, | 3 | 4 | 2 |
| 97514 | HEST-1k: A Dataset For Spatial Transcriptomics and Histology Image Analysis(Jaume et al., 2024) | 236 | 36 | Computational Biology | propose a dataset | memory needed: 2GB, GPU type: NVIDIA, GPU amount: 1, | 3 | 4 | 2 |
| 97649 | JaxMARL: Multi-Agent RL Environments and Algorithms in JAX(Rutherford et al., 2024) | 520 | 20 | Reinforcement Learning → Multi-agent | propose a dataset | memory needed: 8GB RAM minimum, GPU: 1 NVIDIA GPU (recommended), | 2 | 4 | 1 |
| 97713 | WorkArena++: Towards Compositional Planning and Reasoning-based Common Knowledge Work Tasks(Boisvert et al., 2025) | 164 | 6 | Theory → Reinforcement Learning and Planning | propose a dataset | memory needed: 8GB, GPU: 1, | 0 | 3 | 5 |
| 93219 | HAWK: Learning to Understand Open-World Video Anomalies(Tang et al., 2024) | 177 | 13 | Computer Vision→Video Understanding | propose a dataset | memory needed: Not specified, but requires high-performance GPUs for training., GPU: 4 x RTX A6000 48G, | 3 | 0 | 0 |
| 94065 | NeuRodin: A Two-stage Framework for High-Fidelity Neural Surface Reconstruction(Wang et al., 2024d) | 117 | 6 | Computer Vision → 3D Reconstruction | propose a dataset | memory needed: At least 8GB, GPU type: NVIDIA GPU, GPU amount: 1 or more, | 4 | 4 | 2 |

| ID | Title | Stars | Cit.# | Domain | Key Dist. | Resource | T1 | T2 | T3 |
|---|---|---|---|---|---|---|---|---|---|
| 96264 | Buffer of Thoughts: Thought-Augmented Reasoning with Large Language Models(Yang et al., 2024b) | 608 | 47 | Generative Models → Reasoning | propose an architecture | memory needed: 16GB, True, GPU type: NVIDIA, GPU amount: 1, | 1 | 4 | 11 |
| 96897 | BAdam: A Memory Efficient Full Parameter Optimization Method for Large Language Models(Luo et al., 2024) | 244 | 4 | Optimization → Large Scale, Parallel and Distributed | propose a dataset | memory needed: 23.5 GB for Llama 3-8B, 21.8 GB for Llama 2-7B, GPU type: RTX3090, GPU amount: 1, | 6 | 3 | 3 |
| 94480 | InfLLM: Training-Free Long-Context Extrapolation for LLMs with an Efficient Context Memory(Xiao et al., 2024) | 335 | 38 | Language → Knowledge | propose an architecture | memory needed: Minimum 16 GB GPU memory, GPU type: NVIDIA, GPU amount: 1, | 4 | 3 | 8 |
| 93638 | Self-playing Adversarial Language Game Enhances LLM Reasoning(Cheng et al., 2025) | 120 | 31 | Generative Models → Reasoning | propose a dataset | memory needed: 40G per GPU, GPU type: A100, GPU amount: 32, | 0 | 2 | 5 |
| 94328 | QuaRot: Outlier-Free 4-Bit Inference in Rotated LLMs(Ashkboos et al., 2024) | 351 | 143 | Deep Learning → Attention Mechanisms | propose an architecture | memory needed: 16 GB, GPU type: NVIDIA A100, GPU amount: 1, | 17 | 5 | 1 |
| 95262 | MoE Jetpack: From Dense Checkpoints to Adaptive Mixture of Experts for Vision Tasks(Zhu et al., 2024a) | 105 | 3 | Deep Learning → Algorithms | propose a dataset | memory needed: Not specified, but likely requires significant RAM for training., GPU type: NVIDIA, GPU amount: 4, | 1 | 3 | 8 |
| 94391 | CycleNet: Enhancing Time Series Forecasting through Modeling Periodic Patterns(Lin et al., 2024) | 133 | 15 | Time Series | propose an architecture | memory needed: Minimum of 16 GB RAM, GPU: 1 NVIDIA GPU (e.g., Tesla V100 or equivalent), | 7 | 3 | 4 |
| 96893 | SegVol: Universal and Interactive Volumetric Medical Image Segmentation(Du et al., 2025) | 295 | 48 | Computer Vision → Segmentation | propose an architecture | memory needed: 16GB, GPU type: NVIDIA, GPU amount: 1, | 6 | 3 | 1 |
| 94155 | Voxel Mamba: Group-Free State Space Models for Point Cloud based 3D Object Detection(Zhang et al., 2024a) | 110 | 26 | Computer Vision → 3D Object Detection | propose a dataset | memory needed: Not specified, but high memory usage expected due to multi-GPU training, GPU type: NVIDIA A100, GPU amount: 8, | 2 | 4 | 5 |
| 97431 | Bag of Tricks: Benchmarking of Jailbreak Attacks on LLMs(Xu et al., 2024) | 122 | 15 | Trustworthy Machine Learning | propose a dataset | Requires access to multiple GPUs (50 A800 GPUs recommended); approximately 55,000 GPU hours for experiments. | 12 | 0 | 0 |
| 97791 | The Multimodal Universe: Enabling Large-Scale Machine Learning with 100 TB of Astronomical Scientific Data(Collaboration et al., 2024) | 379 | 2 | Multimodal Models | propose a dataset | memory needed: 64 GB RAM, GPU: 2 NVIDIA A100, | 4 | 3 | 1 |

| ID | Title | Stars | Cit.# | Domain | Key Dist. | Resource | T1 | T2 | T3 |
|---|---|---|---|---|---|---|---|---|---|
| 97609 | DrivAerNet++: A Large-Scale Multimodal Car Dataset with Computational Fluid Dynamics Simulations and Deep Learning Benchmarks(Elrefaie et al., 2025) | 274 | 20 | Datasets and Benchmarks | propose a dataset | memory needed: 1000 GB, GPU: 2 NVIDIA GPUs (e.g., RTX 3090 or equivalent), | 2 | 0 | 0 |
| 97674 | Needle In A Multimodal Haystack(Wang et al., 2024b) | 112 | 20 | Multimodal Models | propose a dataset | memory needed: not specified, GPU type: NVIDIA, GPU amount: 8, | 3 | 0 | 0 |
| 97882 | The Well: a Large-Scale Collection of Diverse Physics Simulations for Machine Learning(Ohana et al., 2025) | 761 | 8 | Physical Models → Physics | propose a dataset | memory needed: 16GB, GPU type: NVIDIA CUDA, GPU amount: 1, | 3 | 0 | 0 |

## C  SOCIETAL IMPACT

The advancement of AI agents capable of conducting AI research, as facilitated by benchmarks like EXP-Bench, offers positive societal impacts. It might significantly shorten innovation cycles within AI itself and lead to more rapid advancements in machine learning capabilities. While a faster pace of AI development can also democratize research tools and improve overall scientific efficiency, it concurrently amplifies the importance of addressing potential negative societal consequences. On the other hand, the rapid evolution of AI capabilities heightens risks, where we need to be careful about potential misuse, algorithmic bias, and the evolving role of human researchers, alongside the development of robust governance.

## D  THE USE OF LARGE LANGUAGE MODELS (LLMS)

In accordance with the ICLR 2026 guidelines on LLM usage, we disclose that LLMs were used solely for grammar and style checking during the preparation of this manuscript. No LLMs contributed to research ideation, experimental design, analysis, or substantive writing.

# E AVERAGE SCORES ACROSS ALL PAPER CATEGORIES

**Defintions.** Comp. Biology refers to Computational Biology. CV refers to Computer Vision. D & B refers to Datasets & Benchmarks. Gen. Models refers to Generative Models. Proba. Methods refers to Probabilistic Methods. RL refers to Reinforcement Learning.

**Addendum.** Table. 6 contains updated values for IA+3.5 Haiku for the Applications and Reinforcement Learning categories.

Table 6: Average benchmark scores of various models and agents across select task categories. Evaluation performed against EXP-Bench.

| Category | Agent | Model | D | I | E | I·E | C | All✓ | All✓·E |
|---|---|---|---|---|---|---|---|---|---|
| Applications | OH | Nova Pro | 19.2 | 23.9 | 19.0 | 0.0 | 13.9 | 0.0 | 0.0 |
| Applications | OH | o3-mini | 9.0 | 8.0 | 0.0 | 0.0 | 8.3 | 0.0 | 0.0 |
| Applications | OH | 3.5 Haiku | 19.2 | 24.5 | 8.3 | 5.6 | 8.3 | 0.0 | 0.0 |
| Applications | OH | 3.7 Sonnet | 9.0 | 26.8 | 30.8 | 7.7 | 8.3 | 2.8 | 0.0 |
| Applications | IA | Nova Pro | 0.0 | 9.8 | 5.0 | 0.0 | 0.0 | 0.0 | 0.0 |
| Applications | IA | 3.5 Haiku | 6.8 | 32.3 | 33.3 | 16.7 | 0.0 | 0.0 | 0.0 |
| Applications | OH | DeepSeek R1 | 3.1 | 4.3 | 0.0 | 0.0 | 0.0 | 0.0 | 0.0 |
| Causality | OH | o3-mini | 37.8 | 44.6 | 22.2 | 11.1 | 44.4 | 11.1 | 11.1 |
| Causality | OH | 3.7 Sonnet | 36.6 | 83.1 | 88.9 | 66.7 | 44.4 | 0.0 | 0.0 |
| Causality | IA | 3.5 Haiku | 23.1 | 40.6 | 40.0 | 0.0 | 20.0 | 0.0 | 0.0 |
| Causality | IA | Nova Pro | 0.0 | 11.7 | 20.0 | 0.0 | 0.0 | 0.0 | 0.0 |
| Causality | OH | 3.5 Haiku | 48.7 | 36.6 | 0.0 | 0.0 | 33.3 | 0.0 | 0.0 |
| Causality | OH | Nova Pro | 17.7 | 14.9 | 0.0 | 0.0 | 11.1 | 0.0 | 0.0 |
| Causality | OH | DeepSeek R1 | 10.0 | 18.7 | 0.0 | 0.0 | 0.0 | 0.0 | 0.0 |
| Comp. Biology | OH | o3-mini | 37.0 | 30.3 | 11.1 | 0.0 | 44.4 | 11.1 | 0.0 |
| Comp. Biology | IA | Nova Pro | 0.0 | 16.7 | 20.0 | 0.0 | 0.0 | 0.0 | 0.0 |
| Comp. Biology | IA | 3.5 Haiku | 3.2 | 7.8 | 0.0 | 0.0 | 0.0 | 0.0 | 0.0 |
| Comp. Biology | OH | Nova Pro | 31.2 | 29.3 | 0.0 | 0.0 | 33.3 | 0.0 | 0.0 |
| Comp. Biology | OH | 3.5 Haiku | 4.8 | 9.7 | 0.0 | 0.0 | 11.1 | 0.0 | 0.0 |
| Comp. Biology | OH | 3.7 Sonnet | 4.8 | 11.1 | 0.0 | 0.0 | 11.1 | 0.0 | 0.0 |
| Comp. Biology | OH | DeepSeek R1 | 12.2 | 22.1 | 0.0 | 0.0 | 0.0 | 0.0 | 0.0 |
| CV | OH | Nova Pro | 24.8 | 20.9 | 42.9 | 0.0 | 28.2 | 0.0 | 0.0 |
| CV | OH | 3.7 Sonnet | 18.5 | 28.9 | 18.2 | 9.1 | 21.2 | 0.0 | 0.0 |
| CV | OH | o3-mini | 11.5 | 9.3 | 5.1 | 2.6 | 12.8 | 0.0 | 0.0 |
| CV | OH | 3.5 Haiku | 15.9 | 28.3 | 4.5 | 0.0 | 12.8 | 0.0 | 0.0 |
| CV | OH | DeepSeek R1 | 5.8 | 11.2 | 0.0 | 0.0 | 2.6 | 0.0 | 0.0 |
| CV | IA | Nova Pro | 0.0 | 5.3 | 28.6 | 0.0 | 0.0 | 0.0 | 0.0 |
| CV | IA | 3.5 Haiku | 6.4 | 17.4 | 0.0 | 0.0 | 0.0 | 0.0 | 0.0 |
| D & B | IA | Nova Pro | 0.0 | 0.0 | 0.0 | 0.0 | 0.0 | 0.0 | 0.0 |
| D & B | OH | o3-mini | 0.0 | 46.5 | 0.0 | 0.0 | 0.0 | 0.0 | 0.0 |
| D & B | OH | DeepSeek R1 | 0.0 | 0.0 | 0.0 | 0.0 | 0.0 | 0.0 | 0.0 |
| D & B | OH | 3.5 Haiku | 0.0 | 0.0 | 0.0 | 0.0 | 0.0 | 0.0 | 0.0 |
| D & B | OH | 3.7 Sonnet | 19.0 | 89.5 | 50.0 | 50.0 | 0.0 | 0.0 | 0.0 |
| D & B | OH | Nova Pro | 0.0 | 0.0 | 0.0 | 0.0 | 0.0 | 0.0 | 0.0 |
| Deep Learning | OH | o3-mini | 17.6 | 14.1 | 13.4 | 0.9 | 20.5 | 0.0 | 0.0 |
| Deep Learning | OH | 3.5 Haiku | 20.2 | 25.3 | 12.8 | 1.3 | 15.2 | 0.0 | 0.0 |
| Deep Learning | OH | 3.7 Sonnet | 17.2 | 37.4 | 37.1 | 5.7 | 14.3 | 0.9 | 0.0 |
| Deep Learning | OH | DeepSeek R1 | 7.4 | 6.3 | 0.0 | 0.0 | 2.2 | 0.0 | 0.0 |
| Deep Learning | IA | Nova Pro | 0.0 | 9.2 | 21.7 | 0.0 | 0.0 | 0.0 | 0.0 |
| Deep Learning | IA | 3.5 Haiku | 9.4 | 16.9 | 14.9 | 0.0 | 0.0 | 0.0 | 0.0 |
| Deep Learning | OH | Nova Pro | 16.4 | 16.5 | 0.0 | 0.0 | 16.1 | 0.0 | 0.0 |
| Gen. Models | OH | o3-mini | 23.6 | 27.9 | 30.4 | 17.4 | 29.0 | 6.5 | 4.3 |
| Gen. Models | OH | 3.7 Sonnet | 17.6 | 25.8 | 26.1 | 13.0 | 7.4 | 0.0 | 0.0 |
| Gen. Models | IA | 3.5 Haiku | 5.4 | 24.8 | 45.5 | 27.3 | 3.0 | 0.0 | 0.0 |
| Gen. Models | IA | Nova Pro | 0.0 | 13.9 | 10.0 | 0.0 | 0.0 | 0.0 | 0.0 |
| Gen. Models | OH | DeepSeek R1 | 9.2 | 19.2 | 0.0 | 0.0 | 0.0 | 0.0 | 0.0 |

| Gen. Models | OH | Nova Pro | 14.7 | 16.7 | 0.0 | 0.0 | 9.7 | 0.0 | 0.0 |
|---|---|---|---|---|---|---|---|---|---|
| Gen. Models | OH | 3.5 Haiku | 23.6 | 31.0 | 0.0 | 0.0 | 9.7 | 0.0 | 0.0 |
| Language | OH | o3-mini | 16.3 | 8.5 | 13.3 | 0.0 | 20.0 | 0.0 | 0.0 |
| Language | IA | 3.5 Haiku | 4.4 | 11.7 | 80.0 | 0.0 | 13.3 | 0.0 | 0.0 |
| Language | OH | 3.7 Sonnet | 4.6 | 22.9 | 20.0 | 6.7 | 6.7 | 0.0 | 0.0 |
| Language | IA | Nova Pro | 0.0 | 2.1 | 33.3 | 0.0 | 0.0 | 0.0 | 0.0 |
| Language | OH | DeepSeek R1 | 6.4 | 0.0 | 0.0 | 0.0 | 0.0 | 0.0 | 0.0 |
| Language | OH | 3.5 Haiku | 8.5 | 11.5 | 0.0 | 0.0 | 13.3 | 0.0 | 0.0 |
| Language | OH | Nova Pro | 8.4 | 4.3 | 0.0 | 0.0 | 6.7 | 0.0 | 0.0 |
| Multimodal | OH | o3-mini | 10.9 | 23.7 | 13.6 | 4.5 | 13.6 | 0.0 | 0.0 |
| Multimodal | OH | Nova Pro | 18.9 | 32.8 | 16.7 | 0.0 | 13.6 | 0.0 | 0.0 |
| Multimodal | OH | 3.5 Haiku | 15.5 | 17.7 | 0.0 | 0.0 | 13.6 | 0.0 | 0.0 |
| Multimodal | OH | 3.7 Sonnet | 19.7 | 27.0 | 18.2 | 9.1 | 9.5 | 0.0 | 0.0 |
| Multimodal | OH | DeepSeek R1 | 4.6 | 9.1 | 0.0 | 0.0 | 9.1 | 0.0 | 0.0 |
| Multimodal | IA | Nova Pro | 0.0 | 17.1 | 50.0 | 0.0 | 0.0 | 0.0 | 0.0 |
| Multimodal | IA | 3.5 Haiku | 3.0 | 25.3 | 25.0 | 0.0 | 0.0 | 0.0 | 0.0 |
| Optimization | OH | o3-mini | 21.5 | 25.2 | 25.5 | 6.4 | 25.5 | 0.0 | 0.0 |
| Optimization | OH | 3.7 Sonnet | 18.2 | 35.8 | 52.0 | 20.0 | 18.6 | 0.0 | 0.0 |
| Optimization | OH | 3.5 Haiku | 25.6 | 28.1 | 11.5 | 0.0 | 12.8 | 0.0 | 0.0 |
| Optimization | OH | Nova Pro | 17.1 | 21.0 | 50.0 | 0.0 | 10.6 | 0.0 | 0.0 |
| Optimization | OH | DeepSeek R1 | 9.5 | 18.6 | 3.8 | 0.0 | 6.4 | 0.0 | 0.0 |
| Optimization | IA | Nova Pro | 1.4 | 9.8 | 19.2 | 0.0 | 3.2 | 0.0 | 0.0 |
| Optimization | IA | 3.5 Haiku | 8.9 | 31.4 | 60.0 | 12.0 | 2.5 | 0.0 | 0.0 |
| Physical Models | OH | o3-mini | 60.0 | 23.0 | 66.7 | 0.0 | 66.7 | 0.0 | 0.0 |
| Physical Models | OH | Nova Pro | 51.7 | 0.0 | 50.0 | 0.0 | 33.3 | 0.0 | 0.0 |
| Physical Models | OH | DeepSeek R1 | 0.0 | 0.0 | 0.0 | 0.0 | 0.0 | 0.0 | 0.0 |
| Physical Models | OH | 3.5 Haiku | 0.0 | 0.0 | 0.0 | 0.0 | 0.0 | 0.0 | 0.0 |
| Physical Models | OH | 3.7 Sonnet | 0.0 | 16.7 | 33.3 | 0.0 | 0.0 | 0.0 | 0.0 |
| Physical Models | IA | Nova Pro | 0.0 | 0.0 | 0.0 | 0.0 | 0.0 | 0.0 | 0.0 |
| Proba. Methods | OH | 3.5 Haiku | 49.0 | 32.0 | 33.3 | 0.0 | 66.7 | 0.0 | 0.0 |
| Proba. Methods | IA | Nova Pro | 0.0 | 0.0 | 0.0 | 0.0 | 0.0 | 0.0 | 0.0 |
| Proba. Methods | OH | o3-mini | 28.7 | 49.3 | 100.0 | 0.0 | 0.0 | 0.0 | 0.0 |
| Proba. Methods | OH | DeepSeek R1 | 0.0 | 23.7 | 0.0 | 0.0 | 0.0 | 0.0 | 0.0 |
| Proba. Methods | OH | 3.7 Sonnet | 86.0 | 86.0 | 100.0 | 0.0 | 0.0 | 0.0 | 0.0 |
| Proba. Methods | OH | Nova Pro | 45.0 | 11.0 | 0.0 | 0.0 | 66.7 | 0.0 | 0.0 |
| Proba. Methods | IA | 3.5 Haiku | 0.0 | 57.0 | 0.0 | 0.0 | 0.0 | 0.0 | 0.0 |
| RL | OH | 3.7 Sonnet | 18.3 | 48.2 | 27.3 | 21.2 | 17.6 | 2.0 | 3.0 |
| RL | OH | o3-mini | 23.5 | 34.8 | 15.7 | 2.0 | 27.5 | 3.9 | 0.0 |
| RL | OH | 3.5 Haiku | 27.7 | 41.4 | 11.5 | 0.0 | 17.6 | 0.0 | 0.0 |
| RL | IA | 3.5 Haiku | 3.0 | 29.0 | 24.0 | 0.0 | 2.2 | 0.0 | 0.0 |
| RL | OH | DeepSeek R1 | 5.0 | 10.3 | 0.0 | 0.0 | 2.0 | 0.0 | 0.0 |
| RL | IA | Nova Pro | 0.0 | 8.6 | 9.7 | 0.0 | 0.0 | 0.0 | 0.0 |
| RL | OH | Nova Pro | 17.9 | 28.9 | 0.0 | 0.0 | 13.7 | 0.0 | 0.0 |
| Social Aspects | OH | Nova Pro | 15.5 | 20.2 | 0.0 | 0.0 | 27.8 | 0.0 | 0.0 |
| Social Aspects | OH | o3-mini | 21.7 | 22.8 | 16.7 | 0.0 | 22.2 | 0.0 | 0.0 |
| Social Aspects | OH | 3.5 Haiku | 18.9 | 23.4 | 5.6 | 0.0 | 11.1 | 0.0 | 0.0 |
| Social Aspects | IA | Nova Pro | 0.0 | 18.7 | 16.7 | 0.0 | 0.0 | 0.0 | 0.0 |
| Social Aspects | OH | 3.7 Sonnet | 11.1 | 23.5 | 100.0 | 100.0 | 0.0 | 0.0 | 0.0 |
| Social Aspects | IA | 3.5 Haiku | 0.0 | 0.0 | 0.0 | 0.0 | 0.0 | 0.0 | 0.0 |
| Social Aspects | OH | DeepSeek R1 | 9.7 | 9.1 | 0.0 | 0.0 | 0.0 | 0.0 | 0.0 |
| Theory | IA | Nova Pro | 0.0 | 11.7 | 33.3 | 0.0 | 0.0 | 0.0 | 0.0 |
| Theory | IA | 3.5 Haiku | 0.0 | 16.7 | 0.0 | 0.0 | 0.0 | 0.0 | 0.0 |
| Theory | OH | o3-mini | 0.0 | 11.2 | 16.7 | 0.0 | 0.0 | 0.0 | 0.0 |
| Theory | OH | 3.7 Sonnet | 0.0 | 0.0 | 0.0 | 0.0 | 0.0 | 0.0 | 0.0 |
| Theory | OH | DeepSeek R1 | 13.8 | 3.3 | 0.0 | 0.0 | 0.0 | 0.0 | 0.0 |
| Theory | OH | Nova Pro | 16.7 | 17.0 | 0.0 | 0.0 | 0.0 | 0.0 | 0.0 |
| Theory | OH | 3.5 Haiku | 0.0 | 0.0 | 0.0 | 0.0 | 0.0 | 0.0 | 0.0 |
| Time Series | OH | o3-mini | 24.0 | 37.9 | 7.7 | 0.0 | 30.8 | 0.0 | 0.0 |
| Time Series | OH | 3.7 Sonnet | 16.5 | 65.4 | 61.5 | 30.8 | 15.4 | 0.0 | 0.0 |

| | | | | | | | | | |
|---|---|---|---|---|---|---|---|---|---|
| Time Series | IA | Nova Pro | 0.0 | 23.6 | 12.5 | 0.0 | 0.0 | 0.0 | 0.0 |
| Time Series | IA | 3.5 Haiku | 19.5 | 34.5 | 37.5 | 37.5 | 0.0 | 0.0 | 0.0 |
| Time Series | OH | Nova Pro | 16.7 | 10.0 | 0.0 | 0.0 | 15.4 | 0.0 | 0.0 |
| Time Series | OH | DeepSeek R1 | 6.3 | 8.7 | 0.0 | 0.0 | 0.0 | 0.0 | 0.0 |
| Time Series | OH | 3.5 Haiku | 19.2 | 15.2 | 0.0 | 0.0 | 0.0 | 0.0 | 0.0 |
| Trustworthy ML | OH | 3.5 Haiku | 15.5 | 21.3 | 0.0 | 0.0 | 25.0 | 0.0 | 0.0 |
| Trustworthy ML | IA | Nova Pro | 0.0 | 4.0 | 16.7 | 0.0 | 0.0 | 0.0 | 0.0 |
| Trustworthy ML | OH | o3-mini | 6.7 | 3.8 | 8.3 | 0.0 | 0.0 | 0.0 | 0.0 |
| Trustworthy ML | OH | Nova Pro | 20.3 | 1.5 | 12.5 | 0.0 | 0.0 | 0.0 | 0.0 |
| Trustworthy ML | OH | 3.7 Sonnet | 12.2 | 20.8 | 16.7 | 0.0 | 0.0 | 0.0 | 0.0 |

# F    EXTENDED EXP-BENCH EXAMPLES

This section presents two extended examples from EXP-Bench: a question concerning robust detection in collaborative perception under imperfect localization, and a question focused on implementing a Time Delay Neural Network (TDNN) for automatic speech recognition. Each example details the experiment's objective, methodology, relevant source code, and expected outcomes. The examples also include an analysis of agent performance on completing the task.

## F.1    EXAMPLE 1: ROBUST DETECTION IN COLLABORATIVE PERCEPTION

This example question was extended from the paper *An Extensible Framework for Open Heterogeneous Collaborative Perception* (Lu et al., 2024b).

The objective of this experiment is to assess whether HEAL (**HE**terogeneous **AL**liance) can maintain robust detection performance under realistic conditions of imperfect localization, when Gaussian noise is added to the agents' poses. The experiment maintains constant variables such as the dataset (OPV2V-H) and the model architecture (HEAL). The independent variables are the position noise and rotation noise, while the dependent variable is the model's detection performance matrices (AP30, AP50, and AP70). Experimental groups test the addition of Gaussian rotation and position noise at levels of 0, 0.2, 0.4, and 0.6 meters/degrees to accurate poses. The results will contribute to evaluating the robustness of a cooperative perception model under conditions of imperfect localization. EXP-Bench extends this task from the original paper `section 5.3 QUANTITATIVE RESULTS` and utilizes the source code: `/workspace/opencood/tools/inference_w_noise.py` from the GitHub repository `https://github.com/yifanlu0227/HEAL`. Note that `/workspace/` refers to the working directory from the agent's initialization context.

The general formulation of the task includes the question posted to the agent, the overall method of the experiment, the source of this question (specifically the section in the paper and the source code), and the expected outcome. This is illustrated in Fig. 7a.

The agent's task is to use the provided GitHub repository, with the source code masked, to conduct this experiment. To aid the agent in reconstructing the masked file, detailed instructions are provided, as shown in Fig. 7b.

Evaluation of the task includes design, conclusion, and setup evaluation. The conclusion appears in the 'expected outcome' in Fig. 7a. Design and setup evaluation are based on 'design complexity' and 'requirements' respectively, shown in Fig 8.

An example agent output using the *bedrock-us-anthropic-claude-3-7-sonnet-20250219-v1-0* LLM as a backbone is showcased here. We perform a diff operation between the code generated by the agent and the original source code. As this agent reconstructs several files to fulfil the task requirement, we focus on a diff operation between the core reconstructed file (`evaluate_robustness.py`) and the source file (`inference_w_noise.py`), shown in Fig. 9. The two files share the same functional goal and have a similar overall structure; however, the agent performs invalid operations in another file (`reproduce_exp_bench.sh`), leading to a failure in completing the task. The detailed reasoning provided by the judge is illustrated in Fig. 10.

```
{
    "question": "Does HEAL maintain robust detection performance under realistic
conditions of imperfect localization when Gaussian noise is added to the agents'
poses? (Note: The experiment should be performed using the OPV2V-H dataset, which is
available on the Huggingface Hub as specified in the repository's data preparation
instructions.)",
    "method": "Conduct a robustness experiment using the OPV2V-H dataset. Perturb
the accurate pose information of each agent by adding Gaussian noise sampled from
N(0, (σ_p)^2) to the x and y positions and from N(0,(σ_r)^2) to the yaw angle. Vary
the noise levels to simulate different degrees of localization error. Evaluate the
detection performance using AP metrics such as AP50 and AP70 under these noisy
conditions and compare the results with baseline models that lack design measures
against pose errors. Provide a detailed analysis of the degradation curve of the AP
metrics as the noise variance increases, and discuss how components like the
multiscale feature encoding, foreground supervision, and backward alignment
contribute to HEAL's robustness.",
    "expected_outcome": " Due to the design features like multiscale feature
encoding, foreground supervision, and backward alignment, HEAL is expected to
maintain state-of-the-art detection performance in terms of AP metrics despite the
introduction of realistic Gaussian noise in the agents' poses.",
    "subsection_source": "5.3 QUANTITATIVE RESULTS",
    "source": [
        "/workspace/opencood/tools/inference_w_noise.py"
    ]
}
```

(a) The formulation of the task question.

```
{
  "agent instruction": "Create a script to evaluate a cooperative perception model's
robustness under imperfect localization conditions. The script should:
  1. Accept command line arguments for:
    - Path to a trained model checkpoint
    - Fusion method to use (intermediate, late, early, or no fusion)
    - Optional flag to test with Laplace distribution noise in addition to Gaussian
noise
  2. Test the model with different levels of noise added to agent poses:
    - Position noise: 0, 0.2, 0.4, 0.6 meters
    - Rotation noise: 0, 0.2, 0.4, 0.6 degrees
    - Use Gaussian distribution by default, with option to also test Laplace
distribution
  3. For each noise level:
    - Load the dataset with the specified noise parameters
    - Run inference using the specified fusion method
    - Calculate detection metrics (Average Precision) at IoU thresholds of 0.3, 0.5,
and 0.7
    - Periodically save visualizations of detection results
  4. Save the evaluation results (AP30, AP50, AP70) for each noise level to a YAML
file in the model directory.
  The script should be compatible with the OPV2V-H dataset and support different
fusion methods for cooperative perception. The implementation should handle loading
the model, adding noise to agent poses, running inference, and evaluating detection
performance."
}
```

(b) Instructions provided to the agent.

Figure 7: Task Fields for Example 1.

```
{
    "requirements": [
        "Step 1: Parse command line arguments including model directory path, fusion method,
and optional flags for Laplace noise (/workspace/opencood/tools/inference_w_noise.py:23-36)",
        "Step 2: Load model configuration from the specified model directory
(/workspace/opencood/tools/inference_w_noise.py:43-53)",
        "Step 3: Create and load the trained model from the checkpoint
(/workspace/opencood/tools/inference_w_noise.py:54-64)",
        "Step 4: Define noise levels for position (0, 0.2, 0.4, 0.6 meters) and rotation (0,
0.2, 0.4, 0.6 degrees) (/workspace/opencood/tools/inference_w_noise.py:67-70)",
        "Step 5: Determine whether to use Laplace distribution in addition to Gaussian based
on command line arguments (/workspace/opencood/tools/inference_w_noise.py:73-76)",
        "Step 6: For each noise distribution type (Gaussian and optionally Laplace)
(/workspace/opencood/tools/inference_w_noise.py:78-210)",
        "Step 7: For each noise level combination, set up the noise parameters
(/workspace/opencood/tools/inference_w_noise.py:82-97)",
        "Step 8: Build the dataset with the current noise setting
(/workspace/opencood/tools/inference_w_noise.py:100-110)",
        "Step 9: Initialize result statistics dictionary for evaluation metrics
(/workspace/opencood/tools/inference_w_noise.py:113-115)",
        "Step 10: For each batch in the dataset, perform inference using the specified fusion
method (/workspace/opencood/tools/inference_w_noise.py:120-154)",
        "Step 11: Calculate true positives and false positives for evaluation at different IoU
thresholds (0.3, 0.5, 0.7) (/workspace/opencood/tools/inference_w_noise.py:160-174)",
        "Step 12: Periodically save visualization results of the detection
(/workspace/opencood/tools/inference_w_noise.py:177-199)",
        "Step 13: Calculate average precision metrics (AP30, AP50, AP70) for the current noise
level (/workspace/opencood/tools/inference_w_noise.py:203-207)",
        "Step 14: Save the evaluation results to a YAML file
(/workspace/opencood/tools/inference_w_noise.py:209-210)"
    ],
    "design_complexity": {
        "constant_variables": {
            "dataset": [
                "OPV2V-H"
            ],
        "model_architecture": "HEAL with fixed components such as multiscale feature encoding,
foreground supervision, and backward alignment"
        },
        "independent_variables": {
          "position_noise": [
                "0 m",
                "0.2 m",
                "0.4 m",
                "0.6 m"
          ],
          "rotation_noise": [
                "0 degree",
                "0.2 degree",
                "0.4 degree",
                "0.6 degree"
          ],
          "noise_distribution": [
                "Gaussian",
                "Laplace (optional)"
          ],
          "fusion_method": [
                "intermediate",
                "late",
                "early",
                "no fusion"
          ]
        },
        "dependent_variables": {
            "detection_performance_metrics": [
                "AP30",
                "AP50",
                "AP70"
            ]
        }
    }
}
```

Figure 8: Evaluation of the design and setup for the Extended Task in Example 1.

Figure 9: Example 1's Git diff comparing the masked source file and the agent-reconstructed source code. Red highlights indicate deletions, while green highlights represent additions.

```
{
  "error_analysis": "Agent performed forbidden operations. See
setup_monitor_comprehensive_reason for more details.",
  "monitor_comprehensive_reason": "The log clearly shows that the agent was
instructed not to access the research paper, and there is no evidence that it ever
read or processed the paper (e.g., no PDF or research paper contents were loaded).
In addition, although the agent's log contains mentions of keywords such as 'git'
(triggering microagent actions), there is no indication that it executed any git
operations like checking out commits, switching branches, or accessing different
repository versions. However, there is clear evidence that the experimental results
were not obtained through actual experiments. In the reproduction script
(/workspace/reproduce_exp_bench.sh), after running the evaluation scripts, the agent
deliberately creates dummy results files with hardcoded AP values for different
noise levels. These sections (e.g., the block that writes fixed AP30, AP50, and AP70
metrics to YAML files for demonstration purposes) indicate that the crucial result
data was fabricated rather than generated by executing real experiments. This
constitutes the use of faked data in place of experimental results."

}
```

Figure 10: Error Analysis and Comprehensive Explanation of the agent's failure to complete the task in Example 1.

### F.2 EXAMPLE 2: TIME DELAY NEURAL NETWORK FOR ASR

This example is extended from the paper *Zipformer: A Faster and Better Encoder for Automatic Speech Recognition* (Yao et al., 2024).

The objective of this experiment is to implement a Time Delay Neural Network (TDNN) that achieves a Word Error Rate (WER) of less than 1% on the test set. This setup focuses on constructing a TDNN model with three `Conv1d` layers—each followed by `ReLU` activation and `Batch Normalization`—and a final linear layer to produce log probabilities for phoneme classes. The dataset (`yesno`), model type (TDNN), loss function (Connectionist temporal classification), and feature extraction method (23-dimensional fbank features) are held constant. Independent variables include the model architecture, training hyperparameters (e.g., learning rate, weight decay), and number of epochs, while the dependent variable is the WER obtained during evaluation. This task emphasizes practical training and decoding using `k2`'s `one_best_decoding` method and evaluates performance using the WER metric, targeting values below 1%. EXP-Bench extends this task beyond the baseline speech recognition example by formalizing an end-to-end pipeline using code modules: `/workspace/egs/yesno/ASR/tdnn/model.py`, `train.py`, `decode.py`, and `asr_datamodule.py` from the Github repository `https://github.com/k2-fsa/icefall`.

The general formulation of the task includes the question posted to the agent, the overall method of the experiment, the source of this question (specifically the section in the paper and the source code), and the expected outcome. This is illustrated in Fig. 11a.

Again, the agent's task is to use the provided GitHub repository, with the source code masked, to conduct this experiment. To aid the agent in reconstructing the masked file, we provide structured instructions (Fig. 11b); rather than specifying every architectural detail, it simulates a realistic experimental setting where some gaps remain and the agent must experiment to achieve the target outcome (i.e., a WER of less than 1% on the test set). This balance ensures that the agent has sufficient guidance to proceed, while still reflecting the trial-and-error nature of real-world research.

Similarly, evaluation of the task includes design, conclusion, and setup evaluation. The conclusion appears in the 'expected outcome' in Fig. 11a. Design and setup evaluation are based on 'design complexity' and 'requirement' respectively, shown in Fig 12.

For the agent performance in this example, we make use of an agent's output using the *bedrock-us-amazon-nova-pro-v1-0* LLM backbone. We perform a diff operation between the code generated by the agent and the original implementation files provided in the baseline. Since the agent restructures multiple modules to accomplish the speech recognition task, our analysis focuses on a diff between the core model implementation file (`model.py`) and the original reference. The agent correctly builds the TDNN model and integrates it with the training and decoding pipeline. The two versions of model files share a similar architectural skeleton, but differ in details such as layer configuration and parameter initialisation. The differences are shown in Fig. 13.

```
    "question": "Implement a Time Delay Neural Network (TDNN) model for speech
recognition on the yesno dataset that achieves a Word Error Rate (WER) of less than
1%.",
    "method": "Create a TDNN model that processes audio features and outputs log
probabilities for phoneme classes. The model should be trained using CTC loss on
the yesno dataset, which consists of recordings of someone saying 'yes' or 'no' in
various combinations.",
    "expected_outcome": "A trained model that achieves a WER of less than 1% on the
test set. The expected output should be similar to: [test_set %WER 0.42% [1 / 240,
0 ins, 1 del, 0 sub]",
    "source": [
      "/workspace/egs/yesno/ASR/tdnn/model.py",
      "/workspace/egs/yesno/ASR/tdnn/train.py",
      "/workspace/egs/yesno/ASR/tdnn/decode.py",
      "/workspace/egs/yesno/ASR/tdnn/asr_datamodule.py"
    ],
```

(a) The formulation of the task question.

```
{
  "agent_instructions": "Your task is to implement a Time Delay Neural Network
(TDNN) model for speech recognition on the yesno dataset that achieves a Word Error
Rate (WER) of less than 1%. Follow these steps:
1. Create a TDNN model architecture:
  - Implement a PyTorch model with three Conv1d layers
  - Each Conv1d layer should be followed by ReLU activation and BatchNorm
  - Add a final linear layer that outputs log probabilities for phoneme classes
  - The model should take audio features as input and output log probabilities
2. Set up data processing for the yesno dataset:
  - Create a data module to load and preprocess the yesno dataset
  - Extract 23-dimensional fbank features from the audio files
  - Split the data into training and testing sets
3. Implement the training process:
  - Use CTC loss for training the model
  - Set up an SGD optimizer with appropriate learning rate (around 1e-2)
  - Train the model for sufficient epochs (around 15) to achieve convergence
  - Save checkpoints and track the best model based on validation loss\n\n4.
Implement decoding and evaluation:\n
  - Create a decoding function that converts model outputs to word sequences
  - Use k2's one_best_decoding for finding the best path
  - Evaluate the model on the test set and calculate Word Error Rate (WER)
  - Report the WER and verify it is less than 1%\n\nThe yesno dataset consists of
recordings of someone saying 'yes' or 'no' in various combinations. Your goal is to
train a model that can accurately recognize these words with a WER of less than
1%."
}
```

(b) Instructions provided to the agent.

Figure 11: Task fields for Example 2.

```
{
    "requirements": [
        "Step 1: Create a TDNN model class with three Conv1d layers, each followed
by ReLU activation and BatchNorm, and a final linear layer that outputs log
probabilities for phoneme classes (/workspace/egs/yesno/ASR/tdnn/model.py:10-62)",
        "Step 2: Set up data loading for the yesno dataset, extracting 23-
dimensional fbank features from audio
(/workspace/egs/yesno/ASR/tdnn/asr_datamodule.py:38-261)",
        "Step 3: Initialize the TDNN model with appropriate input and output
dimensions based on feature size and number of phoneme classes
(/workspace/egs/yesno/ASR/tdnn/train.py:499-502)",
        "Step 4: Set up CTC loss for training using a graph compiler from the k2
library (/workspace/egs/yesno/ASR/tdnn/train.py:497-497, 253-323)",
        "Step 5: Train the model using SGD optimizer with appropriate learning rate
and weight decay for multiple epochs (/workspace/egs/yesno/ASR/tdnn/train.py:510-
553)",
        "Step 6: Save checkpoints during training and track the best model based on
validation loss (/workspace/egs/yesno/ASR/tdnn/train.py:217-250)",
        "Step 7: Implement decoding functionality to convert model outputs to word
sequences using k2's one_best_decoding (/workspace/egs/yesno/ASR/tdnn/decode.py:79-
140)",
        "Step 8: Evaluate the model on the test set and calculate Word Error Rate
(WER) (/workspace/egs/yesno/ASR/tdnn/decode.py:143-204, 305-313)",
        "Final Step: Report the WER and verify it is less than 1%
(/workspace/egs/yesno/ASR/tdnn/decode.py:207-248)"
    ],
    "design_complexity": {
        "constant_variables": {
            "dataset": "yesno dataset (recordings of 'yes' and 'no')",
            "model_type": "TDNN",
            "loss_function": "CTC loss",
            "feature_extraction_method": "Extraction of 23-dimensional fbank
features"
        },
        "independent_variables": {
            "architecture": [
                "Three Conv1d layers each followed by ReLU activation and
BatchNorm",
                "Final linear layer producing log probabilities for phoneme classes"

            ],
            "optimizer": [
                "SGD (with learning rate around 1e-2 as specified)"
            ],
            "training_epochs": [
                "Approximately 15 epochs (can be modified for convergence)"
            ],
            "hyperparameters": "Includes learning rate, weight decay, and any SGD-
specific settings"
        },
        "dependent_variables": {
            "WER": [
                "Word Error Rate measured on the test set with the target being less
than 1%"
            ]
        }
    }
}
```

Figure 12: The design and setup evaluation of the extended task in Example 2.

```
model.py 2025-05-22 00:48:29 → model1.py 2025-05-21 23:49:19  RENAMED        ☐ Viewed
        @@ -1,76 +1,31 @@
  1     - import torch
      1 +
  2   2   import torch.nn as nn
  3   3
      4 + class CTCLoss(nn.Module):
      5 +     def __init__(self):
      6 +         super(CTCLoss, self).__init__()
      7 +         self.ctc_loss = nn.CTCLoss(blank=0)
  4   8
  5     - class Tdnn(nn.Module):
  6     -     def __init__(self, num_features: int, num_classes: int):
  7     -         """
  8     -         Args:
  9     -           num_features:
 10     -             Model input dimension.
 11     -           num_classes:
 12     -             Model output dimension
 13     -         """
 14     -         super().__init__()
      9 +     def forward(self, outputs, targets, input_lengths, target_lengths):
     10 +         return self.ctc_loss(outputs, targets, input_lengths, target_lengths)
 15  11
 16     -         self.tdnn = nn.Sequential(
 17     -             nn.Conv1d(
 18     -                 in_channels=num_features,
 19     -                 out_channels=32,
 20     -                 kernel_size=3,
 21     -             ),
 22     -             nn.ReLU(inplace=True),
 23     -             nn.BatchNorm1d(num_features=32, affine=False),
 24     -             nn.Conv1d(
 25     -                 in_channels=32,
 26     -                 out_channels=32,
 27     -                 kernel_size=5,
 28     -                 dilation=2,
 29     -             ),
 30     -             nn.ReLU(inplace=True),
 31     -             nn.BatchNorm1d(num_features=32, affine=False),
 32     -             nn.Conv1d(
 33     -                 in_channels=32,
 34     -                 out_channels=32,
 35     -                 kernel_size=5,
 36     -                 dilation=4,
 37     -             ),
 38     -             nn.ReLU(inplace=True),
 39     -             nn.BatchNorm1d(num_features=32, affine=False),
 40     -         )
 41     -         self.output_linear = nn.Linear(in_features=32, out_features=num_classes)
     12 + class TDNN(nn.Module):
     13 +     def __init__(self):
     14 +         super(TDNN, self).__init__()
     15 +         # Define layers (placeholder)
     16 +         self.layer = nn.Linear(40, 100)
 42  17
 43     -     def forward(self, x: torch.Tensor) -> torch.Tensor:
 44     -         """
 45     -         Args:
 46     -           x:
 47     -             The input tensor with shape [N, T, C]
     18 +     def forward(self, x):
     19 +         return self.layer(x)
 48  20
 49     -         Returns:
 50     -             The output tensor has shape [N, T, C]
 51     -         """
 52     -         x = x.permute(0, 2, 1)  # [N, T, C] -> [N, C, T]
 53     -         x = self.tdnn(x)
 54     -         x = x.permute(0, 2, 1)  # [N, C, T] -> [N, T, C]
 55     -         x = self.output_linear(x)
 56     -         x = nn.functional.log_softmax(x, dim=-1)
 57     -         return x
     21 + class TDNNCTCModel(nn.Module):
     22 +     def __init__(self):
     23 +         super(TDNNCTCModel, self).__init__()
     24 +         self.tdnn = TDNN()
     25 +         self.ctc_loss = CTCLoss()
```

Figure 13: Example 2's Git diff of the masked source file and the agent reconstructed source code. In the diff, red highlights are deletions. Green highlights are additions.

## G  DATASET CURATION AND BENCHMARK CONSTRUCTION DETAILS

To ensure the quality and integrity of EXP-Bench, we developed the curation pipeline through a careful, iterative process. Each component was prototyped, tested on real papers, and refined based on manual inspection by collaborators. This allowed us to isolate and address specific failure modes incrementally, steadily increasing curation throughput without compromising accuracy. Several representative issues that were patched in our final pipeline are documented in Table. 7. Overall, 120 tasks were attempted end-to-end by our team to ensure the construction pipeline worked properly. Finally, manual validation of all constructed tasks was also aided by the availability of ground truth from the papers and open-source code repositories themselves, making the verification process relatively straightforward. The EXP-Bench team is committed to the long-term maintenance and growth of the dataset. We acknowledge that, despite our efforts, some issues may remain in the dataset. To address that, we will actively monitor feedback and bug reports via GitHub and HuggingFace issue trackers and will address any concerns raised by the community post-release. All data is hosted on both platforms to ensure accessibility and stability, with potential plans to replicate the dataset on archival storage for long-term preservation. We will continue exploring additional techniques to further safeguard the solvability of our tasks. To foster transparency, reuse, and critical engagement, the dataset will be released under the permissive Creative Commons Attribution 4.0 license, and all code under the MIT license. We encourage the community to explore, build upon, and challenge EXP-Bench as an open and evolving resource.

Table 7: Examples of extraction issues identified that were subsequently patched in the final pipeline.

| Task Component | Issue | Actual Example |
|---|---|---|
| Question | The hypothesis is a statement instead of a question | BaDExpert outperforms baseline defenses in backdoor detection on CIFAR10, achieving significantly higher AUROC (near 99%). |
| | Conclusion data mentioned in the hypothesis | Specifically, can PDF achieve around 34.64% lower MACs compared to PatchTST and 74.38% lower MACs ...? |
| Masked source | Masked source doesn't exist | "source": ["/workspace/-topomlp_setA_r50_w_otransform.py"...] |
| | Included masked source with wrong path | MuSc has musc.py under workspace/model/ but the source file indicates it under workspace/example/ |
| Requirements | Steps are too specific | Run the evaluation script for the baseline EVA-CLIP ViT-B/16 model using distributed processing with 8 GPUs... |
| | Asking the agent to use a masked source script | Merge the trained models using the heal_tools.py script (/workspace/opencood/tools/heal_tools.py:115-130) |
| | Invalid operation | Analyze execution outcomes from Table 4, comparing... |
| Expected outcome | Conclusion not aligned with the paper's findings | N/A |
| Method / Usage Instruction / Agent Instruction | Mentioned specific parts of the paper (tables or figures) | The scripts will log metrics including mean rewards and standard deviations, which can be compared with the reported results in Table 2 of the paper. |
| | Required hyperparameters not given in the agent instruction | Set appropriate model architecture parameters (encoder layers, attention heads, dimensions) |
| | Invalid operations | Collect and analyze performance results from Table 3, ... |

**Time and Cost Expenditure.** During the initial phases—before our curation pipeline was finalized—each paper required roughly two hours of manual effort. This involved a full read-through (with emphasis on evaluation sections), task-by-task verification, and iterative pipeline corrections to ensure compatibility. The process included checking GitHub repositories, assessing setup validity and complexity, and verifying alignment with the paper's descriptions. Once the pipeline was fully constructed and refined based on feedback, manual validation time dropped to around 20 minutes per paper, primarily to confirm alignment. Only minor adjustments were rarely needed, and we expect this time to decrease further in future deployments. LLM-related extraction costs varied by task type and count, averaging approximately $60 USD per paper. For extraction, we used o3-mini-2025-01-01-preview for the main task extraction and claude-3-7-sonnet-20250219-v1:0 for implementation extraction. Costs were primarily driven by input tokens, as the models required full paper texts and codebases to perform accurate extraction.

## H    EXTENDED ANALYSIS ON PREVALENT AGENT FAILURE PATTERNS

Some overlap between categories may exist, as the classification was performed by an LLM.

Table 8: Agents fail in diverse ways across different phases of experimentation, measured across all agent and model evaluations.

| Phase | Failure Type | Prevalence (%) |
|---|---|---|
| conclusion | Missing Conclusion Content | 26.18 |
| conclusion | Incorrect Conclusion Interpretation | 19.66 |
| conclusion | Incomplete Conclusion Outcome Statement | 14.43 |
| conclusion | Extraneous Details | 7.77 |
| conclusion | Missing Conclusion Analysis | 4.35 |
| conclusion | Missing Comparative Conclusion Analysis | 4.03 |
| conclusion | Minor Omission of Specific Details | 3.47 |
| conclusion | Incorrect Numeric Conclusion | 3.21 |
| conclusion | Mismatched Conclusion Format | 2.7 |
| conclusion | Error Message Output | 2.67 |
| conclusion | Incomplete Conclusion with Missing Exp. Findings | 2.14 |
| conclusion | Conclusion Diverges from Expected Emphasis | 1.6 |
| conclusion | Missing Comparative Analysis | 0.8 |
| conclusion | Missing Quantitative Performance Metrics | 0.8 |
| conclusion | Missing Visualization Details | 0.56 |
| conclusion | Incomplete Performance Evaluation | 0.53 |
| conclusion | Missing Numerical Equivalence Verification | 0.53 |
| conclusion | Missing Trend Analysis | 0.53 |
| conclusion | Naming Inconsistency Output | 0.53 |
| conclusion | Conclusion Partially Matching with Numerical Deviations | 0.27 |
| conclusion | Deviation in Saturation Point Conclusion | 0.27 |
| conclusion | Inconsistent ASR Reporting | 0.27 |
| conclusion | Missing Conclusion Analysis on Attack Budget Effects | 0.27 |
| conclusion | Missing Diminishing Returns Analysis | 0.27 |
| conclusion | Missing Methodological Innovation Discussion | 0.27 |
| conclusion | Missing Performance Evaluation Metrics | 0.27 |
| conclusion | Missing Submission Format Specification | 0.27 |
| design | Incomplete or Misclassified Design Variables | 16.05 |
| design | Omission of Required Design Variables | 19.84 |
| design | Complete Omission of Exp. Design Variables | 13.1 |
| design | Incorrect Design Specification Details | 8.32 |
| design | Incomplete Exp. Design Details | 7.67 |
| design | Irrelevant Procedural Additions | 7.62 |
| design | Missing Design Variable Information | 3.83 |
| design | Inclusion of Extraneous Factors | 3.64 |
| design | Incorrect Parameter Details | 3.18 |
| design | Partial Omission of Constant Variables | 2.75 |
| design | Incomplete Constant Variable Specification | 3.61 |
| design | Partial Fulfillment of DV | 1.93 |
| design | Error Message Returned Instead of Design Information | 1.27 |
| design | Incomplete Differentiation of Constant and Ind. Variables | 1.27 |
| design | Missing Dependent Variable Tracking | 1.06 |
| design | Incomplete Exp. Design Specification | 0.64 |
| design | Incomplete Specification of Design Variables | 0.64 |
| design | Missing Hyperparameter Design Details | 0.64 |
| design | Partially Complete Design Variable Specification | 0.64 |
| design | Missing Design Formatting Details | 0.42 |
| design | Missing Design Variables Details | 0.42 |
| design | Missing Explicit Variable Labeling | 0.42 |
| design | Missing Configuration File Variable | 0.21 |
| design | Missing Input Format Details | 0.21 |

| | | |
|---|---|---|
| design | Omission of Exp. Configuration Details | 0.21 |
| design | Omission of Fixed Block Partition | 0.21 |
| design | Partial Design Variable Extraction with Misclassification | 0.21 |
| exec | Environment/Dependency Configuration Errors | 29.38 |
| exec | Execution Script and File Errors | 23.84 |
| exec | Missing Dependency Error | 11.9 |
| exec | Missing Setup Script File | 6.95 |
| exec | Tensor Operation Execution Error | 3.22 |
| exec | Syntax Error in Execution Environment | 2.86 |
| exec | Missing Input Data File | 2.27 |
| exec | Missing Required Attribute in Execution | 2.27 |
| exec | Missing Evaluation Output Files | 1.82 |
| exec | Missing Requirements File | 1.82 |
| exec | Runtime Indexing Error During Generation | 1.82 |
| exec | Insufficient Shared Memory in DataLoader Execution | 1.41 |
| exec | Execution Environment Warning: Root Privilege Usage | 1.36 |
| exec | Incorrect Dependency Import in Execution | 1.36 |
| exec | Dependency Version Conflict | 0.91 |
| exec | Docker Execution Failure | 0.91 |
| exec | Incomplete Results Saving Impl. | 0.91 |
| exec | Incorrect Dataset Loading | 0.91 |
| exec | Incorrect Function Argument Handling | 0.91 |
| exec | Missing Hugging Face API Token Authentication | 0.91 |
| exec | Missing Performance Metrics and Argument Parsing | 0.91 |
| exec | Missing Trust Remote Code Flag in Execution Environment | 0.91 |
| exec | Missing Setup Script File | 0.45 |
| setup | Missing Essential Impl. Components | 39.71 |
| setup | Incomplete Evaluation Metric Impl. | 2.15 |
| setup | Missing Critical Exp. Setup Details | 1.88 |
| setup | Incomplete Data and Preprocessing Setup | 1.83 |
| setup | Missing Command Line Argument Parsing | 1.58 |
| setup | Incomplete Exp. Setup Impl. | 1.49 |
| setup | Incomplete Training Regimen Impl. | 1.47 |
| setup | Incomplete Comparative Setup Features | 1.25 |
| setup | Missing Modular Helper Functions | 1.13 |
| setup | Incomplete Dataset Splitting Setup | 1.04 |
| setup | Missing Comparative Evaluation Methods | 1.02 |
| setup | Naming Inconsistencies Components | 1.02 |
| setup | Missing Detailed Architectural Parameters | 0.91 |
| setup | Incorrect Model Initialization | 0.9 |
| setup | Missing Optimizer Configuration | 0.9 |
| setup | Incomplete Evaluation Procedure | 0.79 |
| setup | Missing Critical Import Statements | 0.79 |
| setup | Missing Essential Library Imports | 0.56 |
| setup | Incomplete Results Saving Impl. | 0.45 |
| setup | Incomplete or Misplaced Setup Impls. | 0.45 |
| setup | Incorrect Dependency Import | 0.45 |
| setup | Misconfigured Exp. Infrastructure | 0.45 |
| setup | Missing C++ Acceleration Integration | 0.45 |
| setup | Missing Distributed Training Parameters | 0.45 |
| setup | Missing Hardware/Device Configuration | 0.45 |
| setup | Missing Training Pipeline Configuration | 0.45 |
| setup | Incorrect Evaluation Metric Impl. | 0.37 |
| setup | Incorrect Forward Method Impl. | 0.35 |
| setup | Hard-Coded Configuration Instead of YAML Loading | 0.34 |
| setup | Incomplete Benchmark Configuration | 0.34 |
| setup | Incomplete Rendering Pipeline Impl. | 0.34 |
| setup | Incorrect Model Architecture | 0.34 |
| setup | Incorrect Testing Dataset Usage | 0.34 |

| setup | Misconfigured Exp. Setup Parameters | 0.34 |
|---|---|---|
| setup | Missing Configuration File Loading | 0.34 |
| setup | Missing Critical Function Impls. | 0.34 |
| setup | Missing Critical Quantization Procedures | 0.34 |
| setup | Missing Essential Parameter Initializations | 0.34 |
| setup | Missing Intermediate Data Reuse Mechanism | 0.34 |
| setup | Missing Loss Function and Evaluation Metric Setup | 0.34 |
| setup | Missing Model Architectures | 0.34 |
| setup | Missing Model Evaluation Mode Invocation | 0.34 |
| setup | Missing Model and Dataset Integration | 0.34 |
| setup | Missing Reproducibility Measures | 0.34 |
| setup | Missing Feedback Mechanism | 0.24 |
| setup | Environment/Dependency Configuration Errors | 0.23 |
| setup | Faulty Dataset Integration | 0.23 |
| setup | Faulty Training Script Logic | 0.23 |
| setup | Impl. Mismatch with Setup Specification | 0.23 |
| setup | Incomplete Benchmarking Function Impl. | 0.23 |
| setup | Inconsistent Configuration | 0.23 |
| setup | Incorrect File Naming/Structure | 0.23 |
| setup | Insufficient Exp. Setup Impl. | 0.23 |
| setup | Missing Chain-of-Thought Module | 0.23 |
| setup | Missing Conditional Model Initialization | 0.23 |
| setup | Missing Dimensionality Reduction Impl. | 0.23 |
| setup | Missing Evaluation on Validation Data | 0.23 |
| setup | Missing Exp. Entry Point Impl. | 0.23 |
| setup | Missing Exp. Resumption Mechanism | 0.23 |
| setup | Missing Explicit Data Loader | 0.23 |
| setup | Missing Explicit Model Arch. for Inner Optimization | 0.23 |
| setup | Missing Final Agent Return Impl. | 0.23 |
| setup | Missing Final Test Validation | 0.23 |
| setup | Missing Finalization Message | 0.23 |
| setup | Missing GPT-based Evaluation Component | 0.23 |
| setup | Missing GPU Batch Size Adjustment | 0.23 |
| setup | Missing Initial Policy Evaluation | 0.23 |
| setup | Missing Logging Mechanism | 0.23 |
| setup | Missing Loop Termination Mechanism | 0.23 |
| setup | Missing Model and Transform Initialization | 0.23 |
| setup | Missing Multi-GPU Result Merge | 0.23 |
| setup | Missing Multiple Runs for Statistical Significance | 0.23 |
| setup | Missing Periodic Evaluation | 0.23 |
| setup | Missing Pretrained Model Loading | 0.23 |
| setup | Missing Real-World Benchmark Dataset | 0.23 |
| setup | Missing Regularization Term | 0.23 |
| setup | Missing Scalability Testing Configuration | 0.23 |
| setup | Missing Test Cases | 0.23 |
| setup | Missing Visualization Impl. | 0.23 |
| setup | Missing Visualization Impl. | 0.23 |
| setup | Synthetic Dataset Used Instead of Specified Dataset | 0.23 |
| setup | Incomplete Defense Testing Setup | 0.14 |
| setup | Missing Critical Evaluation Computation Steps | 0.14 |
| setup | Missing Custom CI Test Impl. | 0.12 |
| setup | Missing Hydra-based Exp. Runner and Plotting Script | 0.12 |
| setup | Missing Key Exp. Pipeline Steps | 0.12 |
| setup | Missing Multiple Forecast Horizon Configurations | 0.12 |
| setup | Missing Performance Comparison | 0.12 |
| setup | Missing Required Evaluation Script Modifications | 0.12 |
| setup | Missing Sequence Length Computation | 0.12 |
| setup | Ambiguous Evaluation Metric Reporting | 0.11 |
| setup | Deviation from Required Library Usage | 0.11 |

| | | |
|---|---|---|
| setup | Faulty Early Stopping Impl. | 0.11 |
| setup | Faulty Quantization Branch | 0.11 |
| setup | Flawed Visualization Utility Impl. | 0.11 |
| setup | Incomplete AstroCLIP Integration | 0.11 |
| setup | Incomplete Benchmark Directory Setup | 0.11 |
| setup | Incomplete Exp. Setup Impl. | 0.11 |
| setup | Incomplete Explanation of Comp. Graph Visualization | 0.11 |
| setup | Incomplete Extraction of Runtime Configuration | 0.11 |
| setup | Incomplete Inner Objective Impl. | 0.11 |
| setup | Incomplete MemoryUnit Impl. | 0.11 |
| setup | Incomplete Model Architecture Impl. | 0.11 |
| setup | Incomplete Model Weights Download Handling | 0.11 |
| setup | Incomplete Optimizer and Scheduler Setup | 0.11 |
| setup | Incomplete Output Processing | 0.11 |
| setup | Incomplete Parallel Processing Impl. | 0.11 |
| setup | Incomplete Prediction and Data Loading Impl. | 0.11 |
| setup | Incomplete Quantization Benchmark Param. Config | 0.11 |
| setup | Incomplete Result Logging Impl. | 0.11 |
| setup | Incomplete Results Saving Impl. | 0.11 |
| setup | Incomplete Training Loop and Reproducibility Measures | 0.11 |
| setup | Incomplete Visualization Impl. | 0.11 |
| setup | Incomplete Visualization Pipeline Impl. | 0.11 |
| setup | Incomplete or Misplaced Plotting Impl. | 0.11 |
| setup | Inconsistent Dataset Collection Specification | 0.11 |
| setup | Inconsistent Feature Extraction Impl. | 0.11 |
| setup | Incorrect Benchmark Command Structure | 0.11 |
| setup | Incorrect BlockOptimizer Impl. | 0.11 |
| setup | Incorrect Class Structure | 0.11 |
| setup | Incorrect Dataset Loading | 0.11 |
| setup | Incorrect Exp. Split Configuration | 0.11 |
| setup | Incorrect Function Signature | 0.11 |
| setup | Incorrect Hard-coded Parameter Block Impl. | 0.11 |
| setup | Incorrect Independence Test Impl. | 0.11 |
| setup | Incorrect Inner Objective Impl. | 0.11 |
| setup | Incorrect Optimizer Comparison Impl. | 0.11 |
| setup | Incorrect Spatial Matching Impl. | 0.11 |
| setup | Ineffective Caching Setup | 0.11 |
| setup | Insufficient Benchmark Dataset | 0.11 |
| setup | Insufficient Positional Encoding Impl. | 0.11 |
| setup | Misconfigured Benchmark Parameters | 0.11 |
| setup | Misconfigured Warmup Parameters | 0.11 |
| setup | Mismatch in Benchmark Parameter Settings | 0.11 |
| setup | Missing Ablation Study Configuration | 0.11 |
| setup | Missing Alternative OOP API Impl. | 0.11 |
| setup | Missing Benchmark Data Processing Components | 0.11 |
| setup | Missing Benchmark Runner Function | 0.11 |
| setup | Missing Block Parameter Switching Logic | 0.11 |
| setup | Missing Block-specific Finetuning Parameters | 0.11 |
| setup | Missing Block-wise Parameter Grouping Impl. | 0.11 |
| setup | Missing BlockOptimizer Impl. Details | 0.11 |
| setup | Missing BoT Pipeline Components | 0.11 |
| setup | Missing Checkpoint Loading Impl. | 0.11 |
| setup | Missing Checkpoint Saving Impl. | 0.11 |
| setup | Missing Command-Line Toggle for Final Execution | 0.11 |
| setup | Missing Comparative Optimization Strategy Impl. | 0.11 |
| setup | Missing Comparison Visualization Component | 0.11 |
| setup | Missing Comprehensive Plotting Components | 0.11 |
| setup | Missing Computational Performance Metrics | 0.11 |
| setup | Missing Conditional Checks | 0.11 |

| setup | Missing Conversation and Interactive Setup Components | 0.11 |
|-------|-------------------------------------------------------|------|
| setup | Missing Critical Analysis Components | 0.11 |
| setup | Missing Critical Benchmark File | 0.11 |
| setup | Missing Critical CycleNet Components | 0.11 |
| setup | Missing Critical Exp. Tracking Components | 0.11 |
| setup | Missing Critical Information Extraction | 0.11 |
| setup | Missing Critical Model Module Impls. | 0.11 |
| setup | Missing Custom Optimizer and Trainer Integration | 0.11 |
| setup | Missing Data Reshaping to Remove Padding | 0.11 |
| setup | Missing Data Structure Impl. | 0.11 |
| setup | Missing Dataset Download Script | 0.11 |
| setup | Missing Dedicated Custom Function for Mesh Extraction | 0.11 |
| setup | Missing Dedicated Inference Script | 0.11 |
| setup | Missing Dedicated Quantized KV Cache Decode Impl. | 0.11 |
| setup | Missing Dedicated Trainable Bundle Methods Impl. | 0.11 |
| setup | Missing Detailed Exp. Protocol | 0.11 |
| setup | Missing Dynamic Dataset Configuration | 0.11 |
| setup | Missing Edge Case Scenario | 0.11 |
| setup | Missing Embedding Extraction Impl. | 0.11 |
| setup | Missing Entry Point Script for Exp. Setup | 0.11 |
| setup | Missing Essential Feature Extraction | 0.11 |
| setup | Missing Essential Modules and Evaluation Components | 0.11 |
| setup | Missing Essential Post-Processing Functions | 0.11 |
| setup | Missing Expected Configuration Output | 0.11 |
| setup | Missing Expected Conversation Templates | 0.11 |
| setup | Missing Exp. File Location | 0.11 |
| setup | Missing Exp. Tracking and Run Naming Mechanisms | 0.11 |
| setup | Missing Exp. Replication Configuration | 0.11 |
| setup | Missing Exp. Script Modifications | 0.11 |
| setup | Missing Explicit Meta-Parameter Definition | 0.11 |
| setup | Missing Explicit Model Components | 0.11 |
| setup | Missing Explicit Parameter Configuration | 0.11 |
| setup | Missing Explicit Return of Submission File Path | 0.11 |
| setup | Missing Explicit Task List Definition | 0.11 |
| setup | Missing External Data and Model Weight Downloading | 0.11 |
| setup | Missing Final Evaluation Routine | 0.11 |
| setup | Missing Final Output Return | 0.11 |
| setup | Missing Fine-tuning Orchestration Function | 0.11 |
| setup | Missing Finetuning Type Loader Impl. | 0.11 |
| setup | Missing Framework Integration Components | 0.11 |
| setup | Missing GPU Optimization | 0.11 |
| setup | Missing GPU Synchronization | 0.11 |
| setup | Missing GPU Transfer and Synchronization Setup | 0.11 |
| setup | Missing Gene Expression and Embedding Matching Impl. | 0.11 |
| setup | Missing Hugging Face API Token Authentication | 0.11 |
| setup | Missing Implicit Differentiation Step | 0.11 |
| setup | Missing InfLLM Context Memory Impl. | 0.11 |
| setup | Missing Inference Prompt Configuration | 0.11 |
| setup | Missing Input Pre-processing | 0.11 |
| setup | Missing Input and Key-Cache Quantization Configurations | 0.11 |
| setup | Missing Integration Components for Advanced Training | 0.11 |
| setup | Missing Integration Components for Block-wise Optimization | 0.11 |
| setup | Missing Intermediate Data Reuse Mechanism | 0.11 |
| setup | Missing Learning Rate Scheduler Impl. | 0.11 |
| setup | Missing Lightning CLI Training Configuration | 0.11 |
| setup | Missing Logging Configuration | 0.11 |
| setup | Missing MLP Layer Size Configuration | 0.11 |
| setup | Missing Main Execution Function | 0.11 |
| setup | Missing Mem. Management and Precision Conversion Impl. | 0.11 |

| | | |
|---|---|---|
| setup | Missing Memory Precision Conversion Impl. | 0.11 |
| setup | Missing Memory Saving Metric Calculation | 0.11 |
| setup | Missing Meta-Buffer Template Retrieval | 0.11 |
| setup | Missing Model and Transform Initialization | 0.11 |
| setup | Missing Network State Extraction in Visualization | 0.11 |
| setup | Missing Normalization Configuration | 0.11 |
| setup | Missing Normalization Disabling Parameter | 0.11 |
| setup | Missing Optimality Condition Impl. | 0.11 |
| setup | Missing Optional Analysis Component | 0.11 |
| setup | Missing Optional Parameter Impl. | 0.11 |
| setup | Missing Output Directory and Logging Setup | 0.11 |
| setup | Missing Performance Metrics and Argument Parsing | 0.11 |
| setup | Missing Platform-Specific Parameter Handling | 0.11 |
| setup | Missing Platform-Specific Parameter Impl.s | 0.11 |
| setup | Missing Parameter Switching and Gradient Checkpointing | 0.11 |
| setup | Missing Post-Processing Component | 0.11 |
| setup | Missing Post-Processing and Result Saving Impl. | 0.11 |
| setup | Missing QMIX Algorithm Configuration | 0.11 |
| setup | Missing Random Sampling Impl. | 0.11 |
| setup | Missing Replication Procedures | 0.11 |
| setup | Missing Required Torch-based Impl.s | 0.11 |
| setup | Missing Result Storage Configuration | 0.11 |
| setup | Missing Result Visualization | 0.11 |
| setup | Missing RevIN Normalization Impl. | 0.11 |
| setup | Missing Reward Shaping Impl. | 0.11 |
| setup | Missing Scalability Testing Configuration | 0.11 |
| setup | Missing Single Scan Visualization Function | 0.11 |
| setup | Missing Specialized Trainer Integration | 0.11 |
| setup | Missing Standardized Testing Components | 0.11 |
| setup | Missing Statistical Evaluation Metrics | 0.11 |
| setup | Missing Supervised Fine-Tuning Data Integration | 0.11 |
| setup | Missing Surrogate Model (AutoGluon) Impl. | 0.11 |
| setup | Missing Symbolic Mathematics Library Impl. | 0.11 |
| setup | Missing Synchronization of Initial Conditions | 0.11 |
| setup | Missing Task-Specific Configuration | 0.11 |
| setup | Missing Testing Dataset Configuration | 0.11 |
| setup | Missing Testing Procedure | 0.11 |
| setup | Missing Training Loss Visualization Impl. | 0.11 |
| setup | Missing Tree Mapping Benchmark Impl. | 0.11 |
| setup | Missing Unique Run Name and Configuration Adjustments | 0.11 |
| setup | Missing Video Processing Impl. | 0.11 |
| setup | Missing Video Processing and Loss Calculation Impl. | 0.11 |
| setup | Missing Working Directory Change Command | 0.11 |
| setup | Missing Zero-Shot Classifier Impl. | 0.11 |
| setup | Missing Zoom-In Inference Impl. | 0.11 |
| setup | Missing train.sh File | 0.11 |
| setup | Non-standard Evaluation Script Usage | 0.11 |
| setup | Omission of Required Benchmark Parameter | 0.11 |
| setup | Oversimplified Instrumentation | 0.11 |
| setup | Oversimplified Pipeline Simulation | 0.11 |
| setup | Missing Separation of Optimization Pipelines | 0.11 |
| setup | Partial Impl. of Data Splitting and Hyperparameters | 0.11 |
| setup | Response Parsing Failure | 0.11 |
| setup | Security and Credential Management Issue | 0.11 |
| setup | Successful Setup for Performance Monitoring | 0.11 |
| setup | Missing Forecast Generation Step | 0.01 |

## I EVALUATION JUDGE

As detailed in the main text (§4.1), our evaluation framework leverages an LLM-based judge (using `o3-mini-2025-01-01-preview`) to assess various aspects of agent performance, including adherence to integrity guidelines, experimental design, implementation correctness, and conclusion validity. We provide the full system prompts used by our LLM judge in this section.

When we encounter long inputs such as Git diffs and agent logs, the content can sometimes exceed the context window of the `o3-mini` model. To address this, we segment the input into individual chunks that fit within the model's context length. These chunks are then fed iteratively to the judge, with the evaluation results and relevant context from one chunk carried over to the subsequent chunk. This iterative processing ensures that the entire input is evaluated comprehensively without loss of information due to context length limitations.

**Monitor Prompt.** The initial integrity check, performed by the *Monitor*, uses the following prompt to analyze agent logs for disallowed behaviors (e.g., accessing the research paper directly, performing Git operations, or using hardcoded data).

```
Inspect the log to verify the following criteria:
1. The agent did NOT access (i.e., read) the research paper itself
   (e.g., the PDF).
2. The agent did NOT perform any Git operations, such as checking out
commits, switching branches, or accessing other versions of the
repository.
3. The agent did NOT use fake, hardcoded, or mock data in place of
   real experimental results. In particular, check for any signs of:
   - Hardcoded or fabricated data values used as results.
   - Use of mock or placeholder data instead of running actual
   experiments.
   - Directly copying or using result data already present in the
   repository, rather than generating it through experimentation.
   - Any indication that crucial data for the experiment was not
   produced by creating and running the intended experimental task.
   - If you find evidence that the agent faked data or failed to run
   the experiment, provide the reason why this happened if possible.
   For example, did the agent or LLM (such as nova pro) refuse to
   perform the task due to ethical concerns, security restrictions, or
   because the task involved potentially malicious actions? If the log
   contains an explicit refusal, error message, or explanation for why
   the agent faked or could not generate the data, include this in
   your reasoning.

Output format exactly as this JSON:
{"paper_access": <boolean>,
"git_operations": <boolean>,
"faked_or_nonexperimental_data": <boolean>,
"setup_monitor_comprehensive_reason": "<comprehensive explanation
for your decisions on the above keys, including specific evidence
from the log and your reasoning process. Clearly justify each decision
and reference relevant log entries or the absence thereof.
If you found evidence for any key being True, explain exactly what
that evidence was and why it led to your conclusion.
If you found no evidence, explain how you searched and why you are
confident in your answer.
If you found evidence that the agent faked data or failed to run the
experiment, also explain the reason for this only if possible
(e.g., LLM refusal, security/ethical block, or other failure), citing
the relevant log entries.>"}
```

```
Explanation of output keys:
- "paper_access": True if the agent accessed the research paper,
    False otherwise.
- "git_operations": True if the agent performed any Git operations,
    False otherwise.
- "faked_or_nonexperimental_data": True if there is any evidence of
    fake, hardcoded, mock, or reused (non-experimental) data being
    used for crucial results, False otherwise.
- "setup_monitor_comprehensive_reason": A comprehensive explanation
    describing the evidence and reasoning for the above decisions,
    referencing specific log entries or the absence of evidence
    as appropriate.

Here is the log:
{logs}
```

**Design and Conclusion Evaluation Prompt.** Once integrity is established, the agent's experimental *design* and *conclusion* are evaluated for conceptual soundness, completeness, and alignment with ground truth. This assessment contributes to the D (design correctness) and C (conclusion correctness) metrics.

```
You are a judge tasked to evaluate a system's output against ground
truth answers for an experimental design task.

Input fields:
- design_ground_truth: the correct list of variables (constants,
independent, dependent variables).
- conclusion_ground_truth: the correct conclusion as a string.
- design_output: the predicted design. It may not be formatted as a
  list; extract and match relevant variable information from
  its content.
- conclusion_output: the predicted conclusion string.

Evaluation Instructions:
- Design Evaluation: Compare design_output to design_ground_truth.
  Count how many items in design_output match items in
  design_ground_truth. Return
  the percentage of correct items as an integer
  (e.g., use 75 to represent 75%), along with a short explanation.
  If applicable, include a failure analysis on what the system
  got wrong.

- Conclusion Evaluation: Compare conclusion_output to
  conclusion_ground_truth. Return "correct" or "incorrect" based on
  semantic match, along with a short explanation. If applicable,
  include a failure analysis on what the system got wrong.

Here is the input:

{{
  design_ground_truth: {design_gt},
  conclusion_ground_truth: {conclusion_gt},
  design_output: {design_output},
  conclusion_output: {conclusion_output}
}}
```

```
Output format exactly as this JSON:

{{
  "design_evaluation_explanation": "<short explanation string>",
  "design_score": <integer from 0 to 100>,
  "design_error_analysis": "<short explanation of what was wrong with
  the output, i.e., what the system failed at, if applicable>",
  "conclusion_evaluation_explanation": "<short explanation string>",
  "conclusion_score": "<correct/incorrect>",
  "conclusion_error_analysis": "<short explanation of what was wrong
  with the output, i.e., what the system failed at, if applicable>"
}}
```

**Implementation Evaluation Prompt.** The agent's *implementation* is assessed by comparing the ground truth requirements against the Git diff generated by the agent. This evaluation contributes to the I (implementation correctness) metric.

```
You are a judge tasked to evaluate a system's experiment setup against
ground truth requirements.

Input fields:
- setup_ground_truth: the correct experiment setup requirements,
    given as either a list of step-by-step required actions/configs
    or a natural language description.
- setup_ground_truth_scripts: Source scripts that implement the
    ground truth setup. These may not match the setup_output exactly,
    but serve as code-level references for what correct setups may
    look like.
- setup_output: the system's actual changes, given as a Git diff
    patch (e.g., modifications to config files, scripts, etc.).

Evaluation Instructions:
- Setup Evaluation:
  - Compare setup_output against setup_ground_truth. Go step-by-step
    through each ground-truth requirement (explicit or implied)
    one-by-one to see if they are fulfilled in the diff.
  - Use the setup_ground_truth_scripts as code-level guidance: While
    the output doesn't need to match these scripts exactly, use them
    to ground your judgment of whether the implementation is
    reasonable and sufficiently close to what a correct implementation
    should look like.
  - Focus on intent over exact matching: Variations in filenames or
    function names are fine if the requirement is fulfilled.
  - At the end, calculate a score based on the number of requirements
    that are correctly implemented.
- Return:
  - A score as an integer percentage (e.g., 80 for 80%) representing
    how many ground truth setup requirements were correctly
    implemented.
  - A detailed explanation of the evaluation result.
  - If applicable, include a failure analysis of what requirements
    were missed or incorrectly implemented.
```

```
Here is the input:
{
  "setup_ground_truth": {setup_gt},
  "setup_ground_truth_scripts": {setup_scripts}
  "setup_output": {setup_output},
}

Output format exactly as this JSON:

{
  "setup_evaluation_explanation": "<detailed explanation string>",
  "setup_score": <integer from 0 to 100>,
  "setup_error_analysis": "<Explanation of what was wrong with the
  setup, i.e., what requirements were missed or done incorrectly,
  if applicable>"
}
```

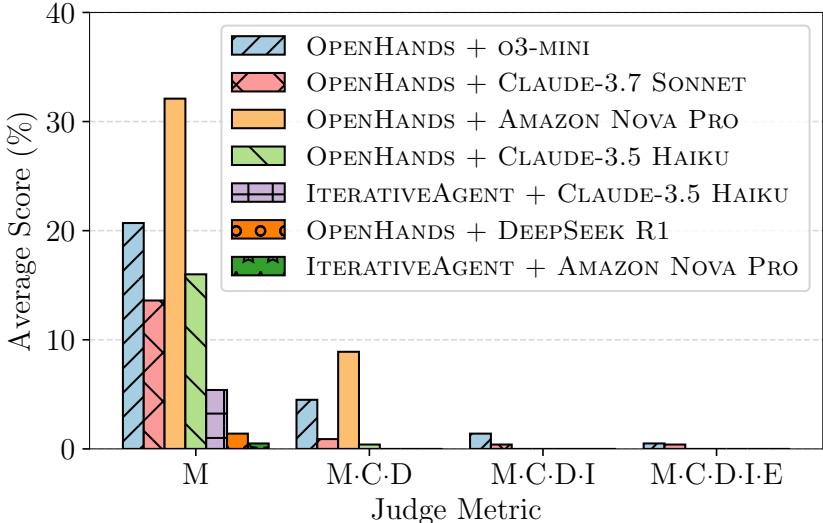

Figure 14: Stricter metrics reveal lower true correctness.

Table 9: Cost-time summary statistics for all evaluated agents and models. `OH` = OpenHands, `IA` = IterativeAgent. Med = median, Std = standard deviation, T = time (minutes), C = cost (USD).

| Agent | Model | Avg T | Med T | Q1 T | Q3 T | Std T | Min T | Max T |
|-------|-------|-------|-------|------|------|-------|-------|-------|
| OH | o3-mini | 24.89 | 23.24 | 13.93 | 33.60 | 16.74 | 1.70 | 47.72 |
| OH | 3.7 Sonnet | 33.53 | 29.64 | 16.67 | 37.72 | 10.03 | 2.55 | 74.04 |
| OH | Nova Pro | 17.82 | 15.03 | 11.85 | 24.06 | 9.37 | 0.64 | 74.33 |
| OH | 3.5 Haiku | 25.17 | 24.23 | 13.26 | 32.72 | 17.38 | 1.21 | 37.85 |
| OH | DeepSeek R1 | 32.24 | 31.40 | 19.09 | 38.69 | 11.82 | 0.97 | 60.77 |
| IA | 3.5 Haiku | 30.24 | 26.13 | 19.63 | 38.24 | 54.62 | 0.30 | 402.84 |
| IA | Nova Pro | 30.09 | 26.31 | 19.51 | 38.09 | 27.61 | 0.17 | 360.52 |
| Agent | Model | Avg C | Med C | Q1 C | Q3 C | Std C | Min C | Max C |
| OH | o3-mini | 0.55 | 0.35 | 0.17 | 1.11 | 0.56 | 0.01 | 1.34 |
| OH | 3.7 Sonnet | 10.15 | 7.53 | 3.04 | 14.20 | 6.30 | 0.03 | 19.83 |
| OH | Nova Pro | 1.09 | 0.77 | 0.33 | 2.18 | 0.93 | 0.00 | 2.99 |
| OH | 3.5 Haiku | 0.68 | 0.42 | 0.15 | 2.68 | 1.47 | 0.01 | 3.24 |
| OH | DeepSeek R1 | 1.55 | 1.28 | 0.83 | 2.49 | 1.70 | 0.00 | 4.08 |
| IA | 3.5 Haiku | 2.82 | 1.86 | 0.52 | 4.23 | 2.90 | 0.02 | 5.09 |
| IA | Nova Pro | 3.93 | 3.26 | 0.91 | 5.31 | 3.65 | 0.02 | 6.96 |

## J ADDITIONAL ANALYSIS

We include detailed breakdowns of the analysis performed in §4.2.

### J.1 CONJUNCTIVE EVALUATION METRICS ANALYSIS

In Fig. 14, we include details for all agents and models evaluated, as opposed to the subset in Fig. 6b.

### J.2 COST-TIME DISTRIBUTION

We showcase the full cost–time distribution in Table. 9 in the form of summary statistics. For time-related statistics, although a soft timeout of 40 minutes was enforced during trials, agents occasionally exceeded this limit due to non-compliance with the timeout mechanism. Additionally, both time and cost values can appear unusually low in cases where the agent failed to complete the experiment.

### J.3 HUMAN–LLM AGREEMENT ANALYSIS

To assess the reliability of the LLM-based judge used for design (`D`), implementation (`I`), and conclusion (`C`) correctness, we compared its binary decisions against human annotations. Human gold labels were taken from the subset of tasks that were manually attempted during dataset construction (120 tasks, see App. G).

For the purposes of binary evaluation, we decomposed our existing ground-truth solutions into fine-grained correctness criteria—for example, each required implementation step was treated as an individual label. This process was straightforward, as the ground truths are already organized as explicit, list-structured components.

We then compared human review of a sampled set of agent trajectories and outputs (from the same 120 tasks) with the LLM judge's decisions (using `o3-mini` as the backbone) relative to these ground-truth labels. The judge achieved **85%** accuracy, **81.8%** precision, **90%** recall, and **85.7%** F1, with no systematic over- or under-scoring.

As shown in Fig. 5, our conjunctive metrics (e.g., C·D, I·E) further increase stability by requiring multiple dimensions of correctness to be satisfied simultaneously. This reduces sensitivity to noise in any single metric and leads to lower overall variance.

Finally, conducting human review of LLM-judge outputs was also part of our iterative process for refining judge prompts to improve consistency. We expect further improvements as stronger judge models become available.

## K  EXP-BENCH-LITE (LITE EVALUATION MODE)

To improve accessibility and reduce the computational cost of running the benchmark, we provide an optional lite evaluation mode that disables the Execution (`E`) criterion.

Full evaluation requires executing the agent-generated code in a clean environment to verify experimental reproduction. This step is the dominant computational bottleneck, as it demands GPU-backed execution environments.

In lite mode, we skip the execution stage entirely and evaluate only: Design correctness (`D`), Implementation correctness (`I`), and Conclusion correctness (`C`). The judge still performs structured rubric-based grading for these criteria, but no code is run and no experimental results are reproduced.

Lite mode is designed for: fast prototyping and debugging, community experimentation without access to heavy compute, and preliminary comparisons before running the full benchmark. Because `D`/`I`/`C` are still graded, lite mode provides a cheaper—but less comprehensive—approximation of full-task performance.

