# OpenReview forum: "EXP-Bench: Can AI Conduct AI Research Experiments?"
_ICLR.cc/2026/Conference — ICLR 2026 Poster_

### Official Review · Reviewer_2fXp · 2025-10-31

**Soundness:** 2
**Presentation:** 2
**Contribution:** 2
**Rating:** 4
**Confidence:** 3

**Summary:**

The authors present a semi-automated pipeline to convert research papers into tasks that evaluate if language models can conduct experiments like that in the original papers. They evaluate several agents on these tasks and present results along with an analysis of failures and successes.

**Strengths:**

* The task production process, barring some caveats below, is indeed fairly scalable which allows for a reasonably large set of tasks.
* The breadth of evaluations and the analysis of failures/successes are strong and valuable additions to the paper.
* There are concerns that since these papers are available on the internet, agents may have memorized approaches to these problems, which significantly reduces the value of this benchmark without further evidence. However, the authors provide analysis that input ablations significantly hurt performance, which does help reduce this risk.

**Weaknesses:**

* The lightweight manual review process means that there are no guarantees that the LLM-generated research questions are not trivial. I additionally have concerns about the validity of the extraction and judging process given the lack of detailed review. Please see the questions section. Without further clarification or evidence, this represents a significant risk that the conclusions of the benchmark are not valid.
* It's fairly restrictive to require the agent use the specified method to answer the objective (and indeed, comparing code diffs and experimental design requires a pretty similar approach). Additionally, the instructions provided to the agent are also extremely specific, which makes the value of the benchmark less useful as perhaps agents could do much better by solving the research question via a different approach.

**Questions:**

* The description of stage 2.1 lacks detail. What are multimodal extraction techniques? Do you use an LLM-based approach?
* I'm confused about how you are guaranteeing that the implementation in stage 2.2 is correct. You first extract the question, method and conclusion and provide this to the agent. The agent then finds a set of scripts from the original codebase and provides a plan of how to execute this to get the conclusion. At this point, is it allowed to modify the scripts? Is it allowed to do so if the script fails and it's asked to refine this? If so, how are you ensuring that there's no trivialization here (i.e. the agent modifies the implementation to make the conclusion obvious or trivial). This is concerning as this is what becomes ground truth.
* How are the execution traces in Stage 3 checked against the expected conclusions? via an LLM judge of some sort? it's not clearly mentioned.
* How could the tasks lack a matched implementation? It seems that your filtering process earlier only allows tasks to pass if Stage 2.2 finds a working implementation. Also, it's not clear if you check that the extracted objective faithfully matches the source paper in cases where you do have an implementation. This is concerning since the implementation was derived from the extracted objective.
* Did you validate that the LLM judge performs accurately by manually labelling a small sample?

---

> ### Author Response · Authors · 2025-11-26
>
> We really appreciate the reviewer’s constructive comments! We now address their questions below.
>
> > It's fairly restrictive to require the agent use the specified method to answer the objective…
>
> Thank you for the observation. Our benchmark is intentionally not designed to test open-ended scientific creativity, though we agree it is an exciting future work! The current goal is to evaluate whether an agent can faithfully carry out a well-specified, paper-grounded experimental procedure—a role closer to a research assistant than an autonomous research innovator. Though this point was already included in our original limitations section, we have now clarified it more explicitly in the revised paper!
>
> > The description of stage 2.1 lacks detail. What are multimodal extraction techniques? Do you use an LLM-based approach?
>
> Thank you for pointing this out. For structured elements such as tables, figures, and headers, we use OCR to locate the visual regions in the PDF and a multimodal LLM to interpret those regions into structured text. We have clarified this multimodal extraction step in the paper.
>
> > I'm confused about how you are guaranteeing that the implementation in stage 2.2 is correct...
>
> Thank you for raising this question. We have now clarified this process more explicitly in the revised paper! The agent is required to reproduce the paper’s stated conclusion. It may modify scripts when necessary, and we prevent trivialization through multiple layers of validation:
>
> (1) Cheating-monitor: We employ an LLM-based monitor during extraction to detect behaviors such as fabricating or mocking results instead of running the actual code.
>
> (2) Clean re-execution: Every proposed implementation is re-executed from scratch inside a clean Docker container. Only tasks that reproduce the expected outcome under this clean environment pass.
>
> (3) Task-paper alignment checks: For all tasks, we manually inspected the extracted task to ensure they faithfully match the source papers and open-source codebases.
>
> (4) Manual cross-checking: We iteratively refined the task construction pipeline through trial-and-error. In doing so, we manually attempted 120 tasks end-to-end (running the code, applying the required changes, and verifying the expected outcomes). This process is detailed in  Appendix G.
>
> > How are the execution traces in Stage 3 checked against the expected conclusions? via an LLM judge of some sort? it's not clearly mentioned.
>
> Thank you for mentioning this. We have added more clarification in our text: in Stage 3, we first re-execute the agent’s reproduction script inside a clean Docker container, ensuring that all dependencies and steps run from scratch. We then use an LLM-based judge to compare these re-executed results against the expected conclusion.
>
> > How could the tasks lack a matched implementation? …
>
> Thank you for pointing this out. We agree the distinction was not sufficiently clear, and we now clarify this more explicitly in the revision. This scenario refers to cases where, in Stage 2.2, the construction agent did not find a directly usable script in the source code and therefore produced an implementation itself. In these cases, the same verification pipeline applies: only implementations that successfully reproduce the paper’s expected conclusion under clean re-execution and manual validation are allowed to pass.
>
> > Did you validate that the LLM judge performs accurately by manually labelling a small sample?
>
> Yes. Specifically, we had manually judged 120 tasks and compared their labels against the LLM judge’s decisions. The judge reached 85% accuracy (81.8% precision, 90% recall, 85.7% F1). We now clarify this procedure in Appendix J.3.
>
> Thank you once again!

---

### Official Review · Reviewer_cvDc · 2025-11-01

**Soundness:** 2
**Presentation:** 4
**Contribution:** 3
**Rating:** 8
**Confidence:** 3

**Summary:**

There are two main contributions of this paper. The first is a semi-automated pipeline for producing such end-to-end AI research tasks based on high-quality conference papers, open-source code implementations, and lightweight human review. The second is a new dataset of 461 tasks built using that pipeline, which claims to be more comprehensive, more "end-to-end", and larger in total size than what exists currently. Several near-frontier models were evaluated using leading open-source agent scaffoldings, and the results show that agents tend to struggle with completely and correctly implementing their plans, as well as with development environment setup and configuration.

**Strengths:**

- building on existing work instead of reinventing everything from scratch. Using existing agent implementations and building on Inspect are doubt-reducing choices.
- The use of a structured workflow and the multi-pass retrieval for generating the tasks—instead of just trying to one-shot LLMs
- Including some amount of human review in the process, validating with a final human validation at the end of the task creation process
- The use of multiple metrics and looking at the distribution of scores across the metrics
- The error analysis was good. It's good to look at the data and explore where exactly agents are failing and what the bottlenecks might be.

**Weaknesses:**

I think the main weakness is that by splitting the paper's attention across the task-generation pipeline and the results of the agents on the tasks, there isn't enough space to deeply explore either. Perhaps my strongest recommendation is to reduce the scope of the paper and focus on either the results of the agents on the tasks--with detailed analysis of agent transcripts, error modes, possible false positives or false negatives--or on the task creation pipeline and validating that these tasks are high quality--especially by having humans attempt them. This may be a terrible suggestion, and it's almost certainly out of scope as a revision for this conference. But if I were to choose the thing that I think would most increase the quality of this paper, it would be to go more focused, more in-depth, and more thorough. This is the main reason for my low "soundness" rating, in that I don't think I have information as it is to _really_ trust the results.

It would be good to report results on each task, not just aggregated results.

**Questions:**

1. Did a human complete any one of the tasks that were generated by the pipeline?
2. How many of the agent traces/trajectories/transcripts were read in full by a human?
    1. How many successful and unsuccessful attempts?
3. Was more open-ended cheating detection—beyond the fixed cheating criteria— conducted?
    1. For example, an LLM-based scatter running on the agent transcripts.
4. What was the purpose of the git diff included in the generated tasks? How was that used, if at all, in scoring?
5. Why was the agent environment created in such a way that the agent had the opportunity to cheat in such an easy way, like reading the paper?
    1. Were any of the runs that were flagged for cheating, and therefore excluded, reviewed to see if perhaps the agent had just misunderstood the instructions?
6. What ability, if any, did the agents have to check the validity of their submissions while working on them?
    1. Were the agents instructed to make sure that their setup scripts could be run in an entirely fresh environment to recreate all dependencies and configuration from scratch?
    2. Were they given an opportunity to test that their script worked?

---

> ### Author Response · Authors · 2025-11-26
>
> We really appreciate the reviewer’s enthusiastic response to our work! We now address their remaining questions.
>
> > Weaknesses..
>
> > How many of the agent traces/trajectories/transcripts were read in full by a human? …
>
> We appreciate the reviewer’s thoughtful suggestions. We agree that a full restructuring of the paper (e.g., focusing exclusively on task-generation or exclusively on agent results) is indeed out of scope for this revision cycle, but we emphasize that one key aspect the reviewer highlighted, "having humans attempt the tasks", is already integral to our pipeline. During task construction, we manually attempted 120 tasks end-to-end (running the code, applying the required changes, and verifying the expected outcomes). Moreover, we cross-checked all tasks (e.g., the question and ground truths) against the source papers and open-source code repositories. This was necessary because we had to have a “stable set” of tasks that we could then use to iteratively refine our task construction pipeline, where the process is outlined in Appendix G.
>
> Second, agent trajectories produced for the same 120 human-validated tasks (including both successful and unsuccessful runs) were also read in full by a human during agent evaluation. We have clarified these in Appendix J.3 in our revised paper.
>
> > Was more open-ended cheating detection—beyond the fixed cheating criteria— conducted?
>
> While reviewing the 120 full agent trajectories for human evaluation, we naturally verified both task correctness and the absence of cheating. We did not observe additional systematic cheating modes beyond those captured by our fixed criteria. Also, our cheating filter is itself an LLM-based monitor that reads the full transcript rather than relying on simple rules, and already detects the broad range of cheating behaviors we would expect.
>
> > What was the purpose of the git diff included in the generated tasks? How was that used, if at all, in scoring?
>
> The git diff records the code and configuration changes the agent made, and is used for post-hoc implementation scoring by comparing these against the ground truth implementation requirements.
>
> > Why was the agent environment created in such a way that the agent had the opportunity to cheat in such an easy way, like reading the paper? …
>
> The opportunity to access the paper comes from the agent scaffolds themselves (e.g., IterativeAgent, OpenHands), which expose file-access tools by default. We preserved these default capabilities to avoid modifying or breaking the frameworks and to evaluate them exactly as they are typically used in prior work. Moreover, we manually reviewed flagged runs throughout the iterative refinement of the LLM-based monitor, using these reviews to verify that the monitor was catching the intended behaviors.
>
> > What ability, if any, did the agents have to check the validity of their submissions while working on them? …
>
> The agents were given substantial freedom, much like a software engineer, to navigate the codebase, write or modify scripts, and run commands. We intentionally kept the instructions open-ended to avoid biasing their workflow, but the agent was required to produce a `reproduce_exp_bench.sh` script, which demanded a setup that could run in a fresh environment with all dependencies and configuration included. The instructions also noted that this script would be used for later verification, giving agents the opportunity—though not an explicit mandate—to check that their setup worked.
>
> Thank you once again for your support!

---

### Official Review · Reviewer_7Rai · 2025-11-01

**Soundness:** 3
**Presentation:** 3
**Contribution:** 3
**Rating:** 6
**Confidence:** 3

**Summary:**

This paper introduces EXP-Bench, a benchmark for evaluating whether AI agents can autonomously perform end-to-end AI research experiments. It is built through a semi-automated pipeline that extracts structured tasks—each including a research question, high-level method, and starter code—from 51 NeurIPS/ICLR 2024 papers, resulting in 461 tasks.
Agents such as OpenHands and IterativeAgent are tested across metrics for design (D), implementation (I), execution (E), and conclusion (C). Results show partial competence (20–35% on sub-tasks) but extremely low full-task success (~0.5%). The work highlights key failure patterns and offers a scalable framework for assessing and improving autonomous research capabilities in LLM-based agents.

**Strengths:**

Originality: The paper introduces a benchmark, EXP-Bench, targeting a rarely studied but crucial problem — evaluating AI agents’ ability to perform complete research experiments. This “end-to-end scientific experimentation” framing goes beyond existing reasoning or coding benchmarks, representing a clear conceptual advancement.
Quality: The proposed semi-automated curation pipeline is technically well-motivated and methodologically sound. It combines multimodal extraction, code analysis, and execution-based validation to ensure high-fidelity, reproducible tasks.
Clarity: The paper is clearly written and well-structured. Figures and tables effectively illustrate the dataset construction process, evaluation metrics, and key failure modes.
Significance: EXP-Bench provides a large-scale and realistic testbed (461 research tasks) for assessing LLM-based research agents.

**Weaknesses:**

1.Reliability of LLM-as-a-Judge evaluation: The benchmark relies exclusively on an LLM-based judge to assess design and conclusion correctness, without any reported human calibration. This raises concerns about evaluation reliability and potential self-consistency bias, since the same modeling paradigm being evaluated also defines the scoring criteria. Including a limited human cross-check or reporting human–LLM agreement statistics would make the results more credible.
2.High computational and API cost.
Running the full benchmark requires substantial API and compute expenses, making it difficult to reproduce or extend. The authors could consider releasing a lightweight subset (similar to SWE-bench Lite or PaperBench Code-Dev) to facilitate community adoption, debugging, and fast prototyping.
3.Lack of resource-aware task curation.
Although the appendix reports per-task hardware and runtime requirements, these resource metrics were not integrated into the benchmark’s task selection pipeline. Incorporating resource-awareness (e.g., filtering by expected GPU hours or memory usage) could improve fairness and scalability when running under standardized Docker environments.

**Questions:**

1.Evaluation strictness vs. creativity:
The benchmark enforces ground-truth matching for design and implementation, but AI agents may propose valid yet creative experimental variants that differ from the reference. How does the evaluation handle such alternative but scientifically sound approaches? Could this strictness penalize genuine innovation?
2.Choice of evaluated models:
Have the authors considered testing open-source, smaller-scale LLMs (e.g., Qwen3) to provide a more accessible baseline for the community? Or do such models perform too poorly to yield meaningful results?

---

> ### Author Response · Authors · 2025-11-26
>
> We are very grateful for the reviewer’s supportive assessment and address the remaining questions below!
>
> > Reliability of LLM-as-a-Judge evaluation…
>
> Thank you for the feedback! We have added a clarification in Appendix J.3 in our revised paper that mentions human–LLM agreement statistics and a limited human cross-check. Specifically, when comparing human review of a subset of agent trajectories/outputs (from 120 tasks) to that of the LLM-judge against the ground truth labels, we found that the judge achieved 85% accuracy, with 81.8% precision, 90% recall, and 85.7% F1, and we observed no systematic over- or under-scoring. Our conjunctive metrics (e.g., C·D, I·E) further stabilize evaluation by requiring multiple dimensions of correctness to hold, resulting in lower variance (Fig. 5). Judge performance also benefited from iterative manual prompt refinement during development, and we expect further gains as stronger judge models become available.
>
> > High computational and API cost…
>
> We agree that the full benchmark is expensive to run. To improve accessibility, we will release a lighter-weight variant similar in methodology to PaperBench Code-Dev. Concretely, we added support for an option to disable the execution (E) criterion in our evaluation pipeline, which is a primary computational bottleneck since it requires executing the code in a clean environment to verify that the results are reproduced. We have included details of this in Appendix K in the revised paper. Moreover, our evaluation pipeline already supports running tasks paper-by-paper, so users do not need to execute the entire benchmark.
>
> > Lack of resource-aware task curation…
>
> Thank you for suggesting this. We have updated our code to include resource-aware filtering by excluding tasks that require more than a configurable number of expected GPUs or GPU memory. The mechanism is modular, allowing users (and future versions of the benchmark) to easily adjust these thresholds.
>
> > Evaluation strictness vs. creativity…
>
> Our benchmark is intentionally not designed to test open-ended scientific creativity, though we agree it is an important future work! The current goal is to evaluate whether an agent can faithfully carry out a well-specified, paper-grounded experimental procedure—a role closer to a research assistant than an autonomous research innovator. Though this point was already included in our original limitations section, we have now clarified it more explicitly in the revised paper!
>
> > Authors considered testing open-source, smaller-scale LLMs (e.g., Qwen3) to provide a more accessible…
>
> Thank you for the suggestion! We plan to include them once we secure additional resources (e.g., having a leaderboard)! With that said, in this version, we did include an open-source model (DeepSeek-R1) to provide such a baseline. At the time of evaluation, even frontier models struggled on these tasks, so we focused our scarce compute on them rather than on smaller LMs that we expected to perform substantially worse.
>
> Thank you again!

---

> > ### Comment · Reviewer_7Rai · 2025-11-28
> >
> > Thanks for the response. After reading the replies, I believe the original rating was reasonable.

---

### Official Review · Reviewer_2DU5 · 2025-11-04

**Soundness:** 3
**Presentation:** 3
**Contribution:** 3
**Rating:** 6
**Confidence:** 4

**Summary:**

The paper contributes a benchmark called EXP-Bench that consists of tasks from 51 papers publised at Computer Science conferences. Tasks include implementing experiments from these papers given a high-level research plan and question.

**Strengths:**

- The paper builds on previous work in evaluating ai agents on coding tasks in ML research well. While other papers have focused on reproducing scientific papers, the tasks in EXP-Bench go one level further and involve implementation of experiments given a high-level outline

**Weaknesses:**

- Missing citations to directly relevant outstanding work [1]
- Number of source papers on which tasks have been generated is rather low.
- Set of chosen models are outdated and do not include latest agentic coding models (GPT-5, Claude Sonnet 4.5, etc.). Even if the contamination problem exists, an analysis of how well models memorize the papers part of the benchmarks could be a good sanity check
- No human error analysis


[1] https://arxiv.org/abs/2409.11363

**Questions:**

- How robust is the LLM-based error analysis when doing human review?

---

> ### Author Response · Authors · 2025-11-26
>
> We sincerely thank the reviewer for their support of our work! We now address their remaining questions.
>
> > Missing citation..
>
> Thank you for pointing that out. We’ve added the citation in the related work section!
>
> > Number of source papers on which tasks have been generated is rather low.
>
> We appreciate the concern. For context, our closest methodological baseline, OpenAI’s PaperBench, is built from 20 papers, whereas our benchmark is a step forward and incorporates 51 papers and produces 461 tasks and 12,737 gradable subtasks. We were able to cover diverse areas of AI research (e.g., CV, NLP, RL; Fig. 3). The current scale reflects a practical balance between coverage and feasibility, which can be expanded further as additional time and compute become available.
>
> > Set of chosen models are outdated and do not include latest agentic coding models (GPT-5, Claude Sonnet 4.5, etc.)...
>
> We agree that including newer models would strengthen the evaluation. At the same time, our primary contribution is the benchmark and methodology; the reported models are intended as representative case studies. Our original selection was constrained by compute and API cost, but we have since run an additional evaluation on GPT-5. We report those results below for completeness:
>
> | Agent      | Model | D    | I    | C    | E    | I·E  | All✓ | All✓·E | #E |
> |------------|-------|------|------|------|------|------|------|--------|-----|
> | OpenHands  | GPT-5 | 26.9  | 23.1 | 4.9  | 24.6 | 8.2  | 0.5  | 1.5    | 234 |
>
> We observe that GPT-5 scores very well on D, but has a middle-of-the-pack score for I, C, and E. One key observation we have is that GPT-5 tends to be more thorough in its analysis, and the agent exhausted its time budget before completing the full setup or reaching the final conclusion.
>
> > How robust is the LLM-based error analysis when doing human review?
>
> When comparing human review of a subset of agent trajectories/outputs (from 120 tasks) to that of the LLM-judge against the ground truth labels, we found that the judge achieved 85% accuracy, with 81.8% precision, 90% recall, and 85.7% F1, and we observed no systematic over- or under-scoring. Our conjunctive metrics (e.g., C·D, I·E) further stabilize evaluation by requiring multiple dimensions of correctness to hold, resulting in lower variance (Fig. 5). Judge performance also benefited from iterative manual prompt refinement during development, and we expect further gains as stronger judge models become available. We have clarified these in Appendix J.3 in our revised paper.
>
> Thank you again!

---

### Meta-Review · Area_Chair_rzNT · 2026-01-07

**Summary:**

This paper introduces EXP-Bench, a benchmark designed to evaluate whether AI agents can execute end-to-end machine learning research experiments derived from published conference papers. Reviewers generally agree that the benchmark targets an important and underexplored capability, going beyond code synthesis to include experimental design, execution, and conclusion validation. The main concerns focus on evaluation reliability (especially LLM-as-a-judge), benchmark scale and representativeness, model coverage, and whether the tasks overly constrain creativity by enforcing paper-faithful reproduction. The rebuttal adds missing citations, GPT-5 results, human–LLM agreement statistics, cost-mitigation options, and clarifications of task construction and validation, substantially strengthening the methodological credibility.

**Reviewer Concerns:**

Reviewer 2DU5’s concerns about missing citations, limited source-paper count, outdated models, and robustness of LLM-based error analysis were fully addressed through added references, scale justification, GPT-5 evaluation, and quantitative human–LLM agreement results.

Reviewer 7Rai’s concerns about LLM-judge reliability, computational cost, resource awareness, evaluation strictness versus creativity, and lack of open-source baselines were largely addressed by human calibration statistics, a lighter benchmark variant, resource-aware filtering, explicit scope clarification, and inclusion of an open-source baseline, with the reviewer explicitly confirming the original rating afterward.

Reviewer cvDc’s concerns about insufficient depth due to broad scope, lack of human task attempts, limited transcript inspection, cheating detection, and scoring clarity were addressed with detailed explanations of human validation on 120 tasks, full trajectory review, cheating-monitor design, and clarification of git-diff usage and execution checks, resolving most methodological doubts.

Reviewer 2fXp’s concerns about task trivialization, restrictive methods, unclear extraction and judging procedures, and insufficient validation were addressed through detailed pipeline clarifications, multi-layer cheating prevention, clean re-execution guarantees, manual alignment checks, and reported human validation of judge accuracy.

**Reviewer Scores:**

Reviewer 2DU5 did not state a score update, but since all concrete technical questions were addressed and additional experiments were added, their concerns appear fully resolved.

Reviewer 7Rai (6) explicitly stated after the rebuttal that the original rating was reasonable, indicating no score change.

Reviewer cvDc did not indicate a score update, but given the detailed clarifications and added human validation, their primary soundness concerns appear largely resolved.

Reviewer 2fXp did not state a score change, but because their methodological and validation questions were directly answered with concrete evidence, most concerns appear resolved, with remaining reservations tied mainly to benchmark scope rather than correctness.

---

### Decision · Program_Chairs · 2026-01-26

Accept (Poster)